# Differential effects of translation inhibitors on *Plasmodium berghei* liver stage parasites

James L McLellan ⬩, Kirsten K Hanson ⬩

**Increasing numbers of antimalarial compounds are being identified that converge mechanistically at inhibition of cytoplasmic translation, regardless of the molecular target or mechanism. A deeper understanding of how their effectiveness as liver stage translation inhibitors relates to their chemoprotective potential could prove useful. Here, we probed that relationship using the *Plasmodium berghei*–HepG2 liver stage infection model. After determining translation inhibition $EC_{50}$s for five compounds, we tested them at equivalent effective concentrations to compare the parasite response to, and recovery from, a brief period of translation inhibition in early schizogony, followed by parasites to 120 h post-infection to assess antiplasmodial effects of the treatment. We show compound-specific heterogeneity in single parasite and population responses to translation inhibitor treatment, with no single metric strongly correlated to the release of hepatic merozoites for all compounds. We also demonstrate that DDD107498 is capable of exerting antiplasmodial effects on translationally arrested liver stage parasites and uncover unexpected growth dynamics during the liver stage. Our results demonstrate that translation inhibition efficacy does not determine antiplasmodial efficacy for these compounds.**

## Introduction

*Plasmodium* parasite resistance to antimalarial drugs is a serious problem throughout endemic regions and underscores the need for new antimalarial drugs that work against novel drug targets (1). In addition to killing asexual blood stage (ABS) parasites, which are solely responsible for the disease, malaria, new compounds capable of killing *Plasmodium* parasites at multiple life cycle stages, including the obligate liver stage of development, are also greatly needed (2). Recently, several compounds were identified with the ability to selectively target parts of the *Plasmodium* cytoplasmic translation machinery, including diverse aminoacyl-tRNA synthetase enzymes (reviewed in references 3, 4) and eukaryotic elongation factor 2 (eEF2) (5), most of which display multistage

antiplasmodial activity. The eEF2 inhibitor, DDD107498 (also known as M-5717 and currently developed as cabamiquine), has successfully completed phase I trials, with demonstrated efficacy against both liver stage and blood stage *Plasmodium falciparum* infections in human volunteers (6, 7). The promise of compounds targeting *Plasmodium* protein synthesis has led to increased efforts to identify additional novel compounds and rational drug targets in the *Plasmodium* cytoplasmic translation apparatus (3, 4, 8, 9, 10, 11). As an increasing number of *Plasmodium*-specific translation inhibitors emerge from these collective efforts, it would be useful to be able to identify compounds with the most desirable activity profiles. Regardless of their molecular target, these compounds converge mechanistically, in that all target the process of cytoplasmic translation. This calls into question the relationship between mechanistic efficacy, how effective such a compound is at inhibiting translation in cellulo, and antiplasmodial efficacy, its ability to kill parasites.

Though most mechanistic antimalarial work has been confined to the ABS, translation inhibitors largely have multistage antimalarial activity, and the demand for protein synthesis during liver stage (LS) development far exceeds that of the ABS; the LS is essentially an amplification step, where a single invading sporozoite will grow and replicate inside a hepatocyte to generate thousands of progeny capable of initiating the blood stage of infection (12). During the ABS, a single merozoite invasion event will generate ~10–32 new erythrocytic merozoite progeny in a single 48-h replication cycle in *P. falciparum* (e.g., references 13, 14), whereas the rodent model parasite *Plasmodium berghei* will generate on average ~12 erythrocytic merozoites in a 24-h cycle (15). In contrast, a single *P. berghei* sporozoite hepatocyte invasion event can generate 1,500–8,000 merozoite progeny during the 2–3 d of LS development, whereas a single *P. falciparum* LS schizont generates 30–40,000 hepatic merozoites during a 6-d growth period ((16) and references therein). Because the biosynthetic output needed to support LS growth and hepatic merozoite production is so much greater than that required for erythrocytic merozoite production, replicating LS parasites might be particularly sensitive to translation inhibition.

Previously, we developed a single-cell quantitative translation assay for *P. berghei* liver stage parasites (17), also known as

Department of Molecular Microbiology and Immunology and STCEID, University of Texas at San Antonio, San Antonio, TX, USA

Correspondence: kirsten.hanson@utsa.edu

exoerythrocytic forms (EEFs), using o-propargyl-puromycin (OPP) (18), to label the LS nascent proteome in cellulo, and showed that liver stage translation inhibition efficacy varied between known translation inhibitors when tested at high, presumably saturating, concentrations. Our estimated translation inhibition $EC_{50}$ for DDD107498 in *P. berghei* LS was 13.4 nM (17), substantially higher than the $EC_{50}$s for antiplasmodial activity reported against *P. berghei* and *Plasmodium yoelii* liver stages and *P. falciparum* ABS, which were 1.65, 0.97, and 1 nM, respectively (5, 19). DDD107498 was also a less efficacious inhibitor of *P. berghei* LS translation at a high, presumably saturating, concentration, than others tested (17). These differences are somewhat surprising, given the extremely strong genetic evidence that *P. falciparum* ABS parasites evolve resistance to DDD107498 in vitro, in mouse models and in humans via mutations in the *P. falciparum* eukaryotic elongation factor 2 (eEF2) gene (5, 6, 20, 21), encoding a highly conserved GTPase required to catalyze translocation during the translation elongation process. Taken together, these differences led us to question whether a translation inhibitor's antiplasmodial efficacy—defined here, with respect to the liver stage, as the ability to prevent the formation and release of hepatic merozoites—is driven by the translation inhibition efficacy of the compound. We thus decided to use the *P. berghei*–HepG2 infection model to probe the relationship between translation inhibition efficacy and antiplasmodial efficacy by examining the effect of a brief period of translation inhibition during LS schizogony on parasite protein synthesis, growth, hepatic merozoite production, and merosome/detached cell release by comparing five mechanistically distinct translation inhibitors tested at equivalent effective concentrations.

## Results

### Translation inhibitor characteristics and potency determination

As our goal was to study the LS parasite response to a brief period of translation inhibition when the translational output was high, we first used OPP labeling of the *P. berghei* LS nascent proteome to confirm our previous result that parasites at 28 h post-infection (hpi) have a greater translational intensity than those at 48 hpi (17). As the average translational intensity at 28 hpi was more than double that at 48 hpi (Fig S1A), we elected to use a 4-h treatment window from 24 to 28 hpi, with nascent proteome labeling from 27.5 to 28 hpi in our experiments. We selected five compounds representing a diverse collection of molecular targets, modes/mechanisms of action, parasite versus host selectivity, and potencies. Four of these compounds were used in our previous work to develop the LS translation assay: anisomycin, bruceantin, DDD107498, and MMV019266. Anisomycin is a pan-eukaryotic elongation inhibitor that binds the ribosomal A-site (22), and has similar activity against *P. berghei* LS translation and HepG2 translation (17). Despite this dual activity and high translation inhibition efficacy, a 24–28 hpi treatment with a high concentration of anisomycin was completely reversible, in terms of parasite translation levels, 20 h after compound washout (17). Bruceantin, a translation elongation inhibitor that binds the ribosomal A-site and

only efficiently inhibits monosomes (23, 24), was both the most potent and most efficacious inhibitor of *P. berghei* LS translation, and also induced the most variable single parasite outcomes at concentrations evoking submaximal effects (17); 4 h of treatment with 3.7 nM bruceantin led to extremely heterogeneous responses in individual parasites, with ~83% of the sampled population translationally inhibited, whereas the remaining ~17% appeared unaffected (Fig S1B). In contrast, parasites treated with the aforementioned DDD107498 maintained a normal population distribution at concentrations evoking both submaximal and maximal effects (Fig S1C). MMV019266, a multistage active Malaria Box compound (25), which was identified in our previous study as a *Plasmodium*-specific translation inhibitor (17), is likely to target the cytoplasmic isoleucyl-tRNA synthetase, as has been demonstrated for several structurally related thienopyrimidines (26). Using a parasite reduction ratio (PRR) assay (27), 10x $EC_{50}$ MMV019266 was shown to kill the majority of *P. falciparum* ABS parasites after 24 h of treatment (26), a rapid rate-of-kill similar to that of the fast-acting reference antimalarial artesunate. This rapid ABS rate-of-kill of MMV019266 is in marked contrast to that of DDD107498, which does not reduce the viable parasite population at all after a 24-h 10x $EC_{50}$ treatment (5), and has a PRR similar to that of atovaquone, the slowest-acting reference antimalarial in the assay. We also chose to test another recently identified parasite-specific aminoacyl-tRNA synthetase (aaRS) inhibitor, LysRS-IN-2, identified in a target-based screen of compounds capable of inhibiting *P. falciparum* cytoplasmic lysyl-RS activity (28); LysRS-IN-2 has an ABS speed-of-kill even slower than that of atovaquone (28). Using a competitive OPP (co-OPP) assay (17), we demonstrate that LysRS-IN-2 is a specific, and likely direct, inhibitor of *P. berghei* LS translation (Fig S1D).

We first determined the translation inhibition potency of each selected inhibitor, tested in 8- or 10-point, threefold serial dilution, with 3.5 h of pre-treatment followed by 30 min of OPP labeling in the presence of the compound of interest (Fig 1A and B; see Table S1 for $EC_{50}$, 95% CI, and R square). As we previously demonstrated (17), the pan-eukaryotic translation inhibitors anisomycin and bruceantin are active against both *P. berghei* and HepG2 translation, but display differences in potency and selectivity. Anisomycin is nearly equipotent against *P. berghei* EEFs and HepG2 cells with $EC_{50}$ values of 262 and 198 nM, respectively (Fig 1A). Both host and parasite show a small concentration-dependent increase in translation initially before switching to translation inhibition, with the entire population of single parasites shifted (Fig 1A and B). Bruceantin is more active against *P. berghei*, with an $EC_{50}$ of 3.72 nM compared with 13.7 nM in HepG2 (Fig 1A). At the single parasite level (Fig 1B), bruceantin appears to be the only compound tested for which a concentration that evoked submaximal translation inhibition shows the parasite population effectively split into a group of strongly responding parasites, the majority, and those that show little, if any, translation inhibition. Of the three parasite-specific translation inhibitors, DDD107498 was the most potent, with a translation inhibition $EC_{50}$ of 11.3 nM, the MMV019266 $EC_{50}$ was 456 nM, and the least potent compound tested was LysRS-IN-2, with an $EC_{50}$ of 6,630 nM (Fig 1A). Canonical LS compound activity assays, with treatment throughout LS development until a 48 hpi readout of parasite biomass, demonstrated that DDD107498 was ~9.4-fold less potent in inhibiting LS translation than in inhibiting LS growth, whereas LysRS-IN-2

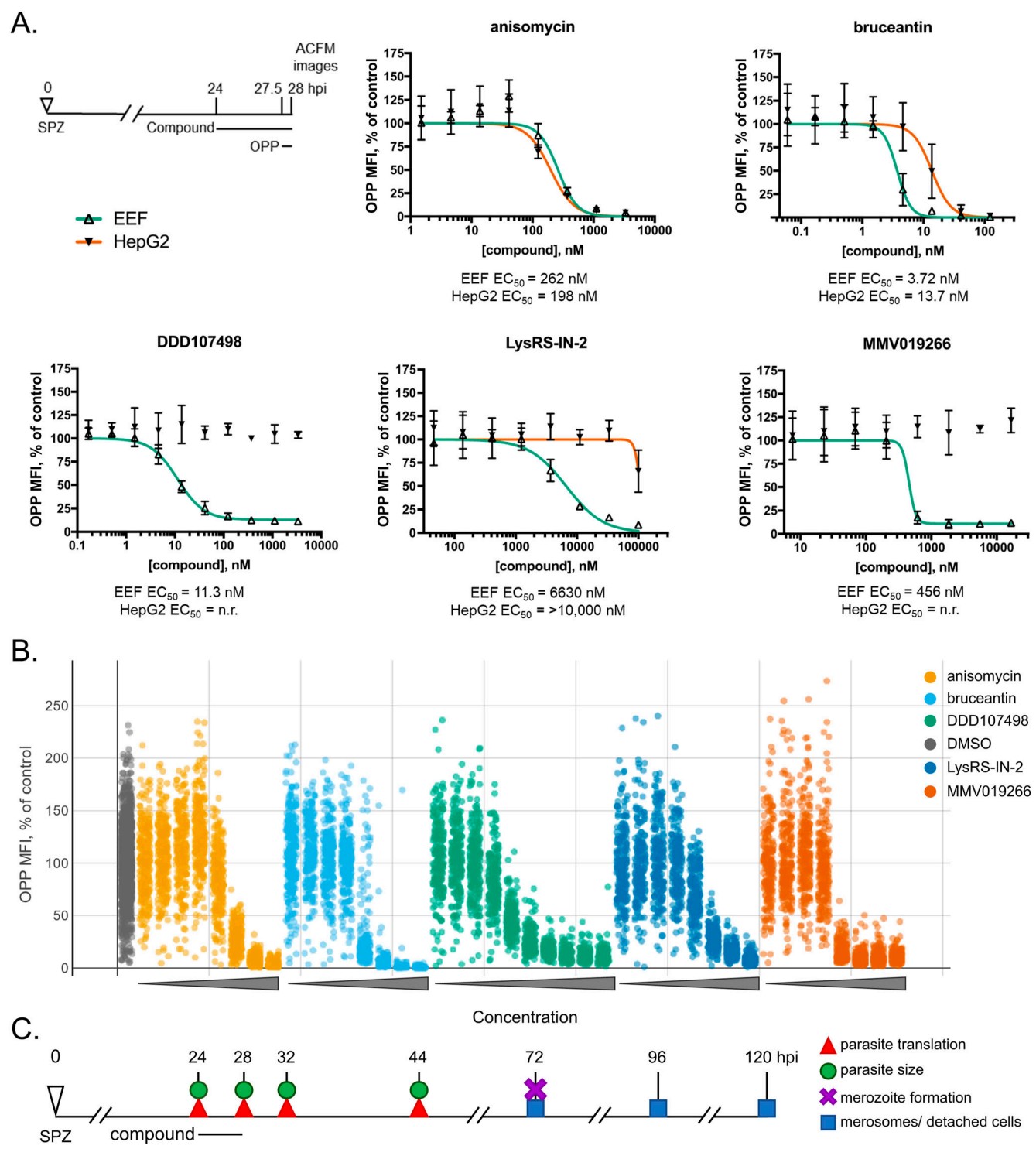

**Figure 1. Translation inhibition potency determination and experimental overview.**
**(A, B)** Translation inhibition was quantified in *P. berghei* exoerythrocytic forms (A, B) and matching in-image HepG2 (A) after acute pre-treatment from 24 to 28 hpi, as shown in the schematic. Compounds were tested in an 8- or a 10-point, threefold serial dilution. **(A)** Concentration–response curves. Single data points represent the mean translation at each concentration, normalized to DMSO controls, and error bars show the SD from n = 3 independent experiments. The absence of line indicates that no curve was fit. **(B)** Single LS parasite concentration–response. For each compound, concentrations are plotted moving from lowest (left) to highest (right) as indicated by the triangles. The top concentrations tested were 3,333 nM for anisomycin and DDD107498, 123 nM for bruceantin, 16,667 nM for MMV019266, and 100,000 nM for LysRS-IN-2. N = 3 independent experiments combined; each dot represents the mean translation intensity (OPP-MFI) in a single parasite normalized to DMSO controls, n = 10,461 exoerythrocytic forms in total. The full concentration–response dataset can be explored via interactive dashboards in our KNIME hub workflow:

and MMV019266 were ~3.6- and ~2.5-fold less potent, respectively (Table S1). DDD107498 had the shallowest slope of the compounds tested, whereas MMV019266 had a markedly steeper slope than the other four compounds (Fig 1A and B). Even though saturating effects on translation were achieved with both DDD107498 and MMV019266, they were noticeably less efficacious than anisomycin and bruceantin, where parasite translation is nearly undetectable (Fig 1A).

To probe the relationship between translation inhibition efficacy and antiplasmodial efficacy in the LS, we designed an experiment to quantify parasite translation and size at the end of a 4-h treatment with 5x and 10x $EC_{50}$ concentrations of each compound, then, after compound washout, to monitor translation and EEF size, along with late liver stage maturation, merozoite formation, and merosome/detached cell release in experimentally matched samples (see Fig 1C for experimental schematic). In addition, DDD107498 was also tested at a lower concentration corresponding to 10x $EC_{50}$ for LS biomass reduction; as this $EC_{50}$ value falls between 1 and 2 nM, we decided to round up, and use a 20 nM concentration, which corresponds to 1.8x our calculated translation inhibition $EC_{50}$. At 24 hpi, pre-treatment controls were OPP-labeled and fixed to determine a baseline for parasite translational output and size, and the remaining samples were compound-treated. At 28 hpi, a subset of the treated samples was OPP-labeled and fixed to quantify translation inhibition and parasite size at the end of the 4-h treatment (Fig 1C). The remaining treated samples were subjected to a stringent compound washout protocol, where the wash volume and total number of media exchanges were calculated to ensure a ≥4-log reduction in compound concentration per well; for example, the concentration in the 10x $EC_{50}$ treatments after washout was reduced to ≤0.001x $EC_{50}$. After the washout, parasite recovery was assessed via both translational intensity and size at 32 hpi (4 h after washout) and 44 hpi (16 h after washout). After processing, individual parasites from the 24, 28, 32, and 44 hpi samples were imaged via automated confocal feedback microscopy (17, 29). At 72 hpi, LS parasite maturation and hepatic merozoite formation were assessed by quantifying the percentage of monolayer EEFs expressing merozoite surface protein 1 (MSP1) and apical membrane antigen 1 (AMA1) in immunolabeled monolayers; we collected the culture medium from the same wells to quantify merozoite-filled, detached HepG2 cells, the presentation of merosome formation in the *P. berghei*–HepG2 infection model (30). In parallel, independent samples were set up for repeated merosome collection every 24 h at 72, 96, and 120 hpi, with complete medium replacement after each collection (Fig 1C). Merosome release in the *P. berghei*–HepG2 infection model is considered largely completed by 65 hpi (30, 31), and the continually replicating HepG2 cells begin to overgrow the monolayer, but we hoped that we could separate potential compound-induced developmental delays, or cytostatic effects, from those that likely represent parasite killing, or cytotoxic effects.

## Translation inhibition efficacy varies between compounds

Translation inhibition efficacy was quantified in image sets generated by automated confocal feedback microscopy (ACFM) in both *P. berghei* liver stage parasites (Fig 2A; see Fig S2A for data reproducibility between experiments) and the in-image HepG2 cells (Fig S3A) at 28 hpi after 4 h of treatment. As expected, parasite translation was significantly inhibited for all compound concentrations tested compared with the DMSO control (one-way ANOVA, $P < 0.0001$). In addition, substantial differences in the ability to inhibit parasite translation were seen between the compounds at both the 5x and 10x $EC_{50}$ concentrations (one-way ANOVA; 5x $EC_{50}$, $P < 0.0001$; 10x $EC_{50}$, $P = 0.002$; for Tukey's multiple comparisons of experimental means, see Table S2). All compounds displayed greater inhibitory activity at 10x $EC_{50}$ than 5x $EC_{50}$, except for MMV019266 and bruceantin where the two concentrations induced essentially equivalent effects (Figs 2A and S2A, Table S2 [A]). MMV019266 and LysRS-IN-2 were the most effective of the parasite-specific inhibitors at 5x and 10x $EC_{50}$, respectively, whereas DDD107498 was the least effective of the parasite-specific translation inhibitors at both effective concentrations (Figs 2A and S2A). MMV019266 and Lys-IN-2 caused >90% translation inhibition at both concentrations, but DDD107498 never reached 90% at either 5x or 10x $EC_{50}$ and resulted in only 64% translation inhibition at 1.8x $EC_{50}$ (Table S2). The two pan-eukaryotic inhibitors bruceantin and anisomycin were highly efficacious; on average, the percent LS translation inhibition by bruceantin was >99% in both 5x and 10x $EC_{50}$ concentrations, whereas anisomycin-induced translation inhibition was ~97% and 99% at 5x and 10x $EC_{50}$, respectively (Figs 2A and S2A and Table S2).

We also quantified parasite size after acute translation inhibition (Fig 2B; see Fig S2B for data reproducibility between experiments). Compared with DMSO controls, significant reductions in parasite growth were seen for all compound concentrations tested (one-way ANOVA, $P < 0.0001$), and significant differences in the growth inhibition were seen between compounds at 5x and 10x $EC_{50}$ concentrations (one-way ANOVA; 5x $EC_{50}$, $P = 0.0006$; 10x $EC_{50}$, $P = 0.0028$; for Tukey's multiple comparisons of experimental means, see Table S2). Because of the causal relationship between protein synthesis and cell growth, we hypothesized that the rank order of compounds for LS growth inhibition at each multiple of $EC_{50}$ would be identical to that of LS translation inhibition, but whereas the most and least effective translation inhibitors correspondingly caused the most and least parasite growth inhibition at both 5x and 10x $EC_{50}$, the other three compounds did not rank equivalently (Fig 2B; see Table S2 for mean values per treatment). Parasite growth was most strongly inhibited in bruceantin and LysRS-IN-2 treatments, where EEF growth inhibition was ~39% and 43% of DMSO controls in the 5x and 10x $EC_{50}$ treatments, respectively, despite bruceantin inducing a significantly stronger effect on translation than LysRS-IN-2 (Fig 2A and B, Table S2). DDD107498 had the least effect on parasite growth during the treatment window, with mean growth inhibition of ~10%, 18%, and 24% in 1.8x, 5x, and 10x $EC_{50}$-treated EEFs, respectively (Table S2).

---

https://hub.knime.com/-/spaces/-/~TZCrKvv3sbJwM_xP/current-state/. **(C)** Experimental setup to probe the effects of acute translation inhibition and the relationship between translation inhibition efficacy and antiplasmodial efficacy with inhibitors tested at equivalent effective concentrations. Source data are available for this figure.

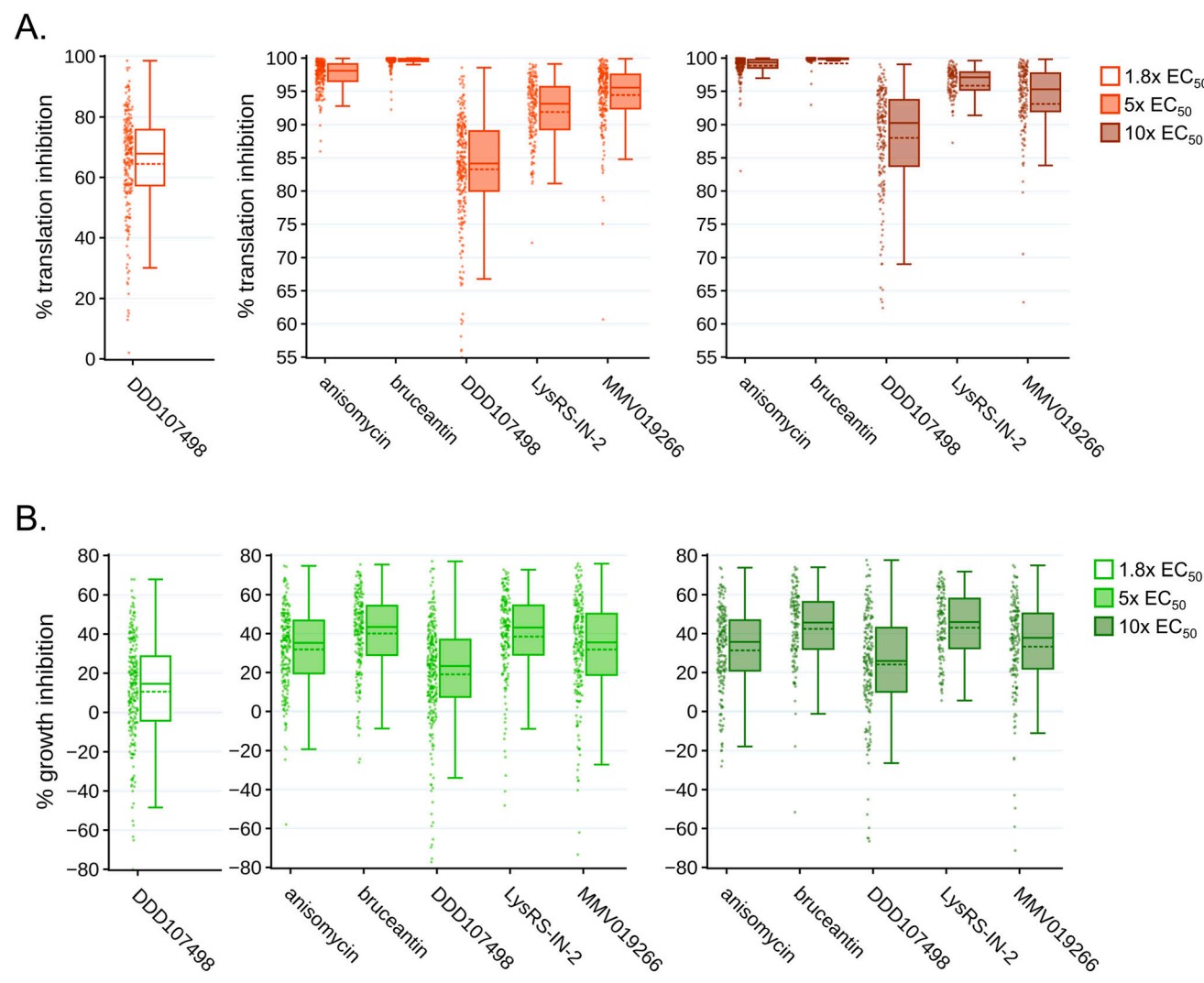

**Figure 2. Differential effects of translation inhibitors tested at equivalent effective concentrations.**
**(A, B)** Translation inhibition (A) and growth inhibition (B) at 28 hpi in *P. berghei* LS, after 4-h acute pre-treatments. Individual data points represent single parasites normalized to in-plate DMSO controls set to 0. Boxplots show combined data from all three independent experiments, with dotted lines reporting treatment means. **(A, B)** OPP-MFIs (A) and parasite areas (B) were normalized to the DMSO control mean, which was set to 100, on a per-experiment basis. Statistical significance was assessed using a one-way ANOVA with Tukey's multiple comparisons testing, reported in Table S2. **(A, B)** To facilitate an easier comparison between treatments, plots were truncated at 55% (A) and −80% (B), removing a total of 8 and 12 individual outlier exoerythrocytic forms, respectively. The full dataset can be explored via interactive dashboards in our KNIME hub workflow: https://hub.knime.com/-/spaces/-/~EcnvMwYtqylu2reV/current-state/.
Source data are available for this figure.

### Translation inhibition efficacy does not determine either translation recovery kinetics or parasite growth after compound washout

Our central hypothesis was that compounds capable of causing the most complete shutdown of parasite translation would result in stronger and more long-lasting effects on *P. berghei* liver stages after compound washout. To our surprise, this was clearly not true for translation recovery, though significant differences in translational output were seen between parasites treated with the different compounds at both 5x and 10x $EC_{50}$ concentrations (ordinary two-way ANOVA; 5x $EC_{50}$, $P$ = <0.0001; 10x $EC_{50}$, $P$ = <0.0001; for Tukey's multiple comparisons, see Table S2). The populations of parasites

treated with anisomycin and LysRS-IN-2 recovered the fastest. 4 h after compound washout, these treatment groups were translating at levels comparable to the DMSO controls and had exceeded the magnitude of control translation by 44 hpi (Fig 3A). EEFs treated with bruceantin and DDD107498, the most and least effective translation inhibitors tested, respectively, showed nearly identical levels of translational output 4 h after compound washout (Fig 3A and Table S2). However, 16 h after washout, bruceantin-treated EEFs had completely recovered and even exceeded the control translational intensity, whereas those treated with DDD107498 showed little, if any, further recovery (Fig 3A). Parasites treated with 1.8x $EC_{50}$ DDD107498 recovered substantially in 4 h to ~67% of control translational intensity, with a near-complete recovery to control

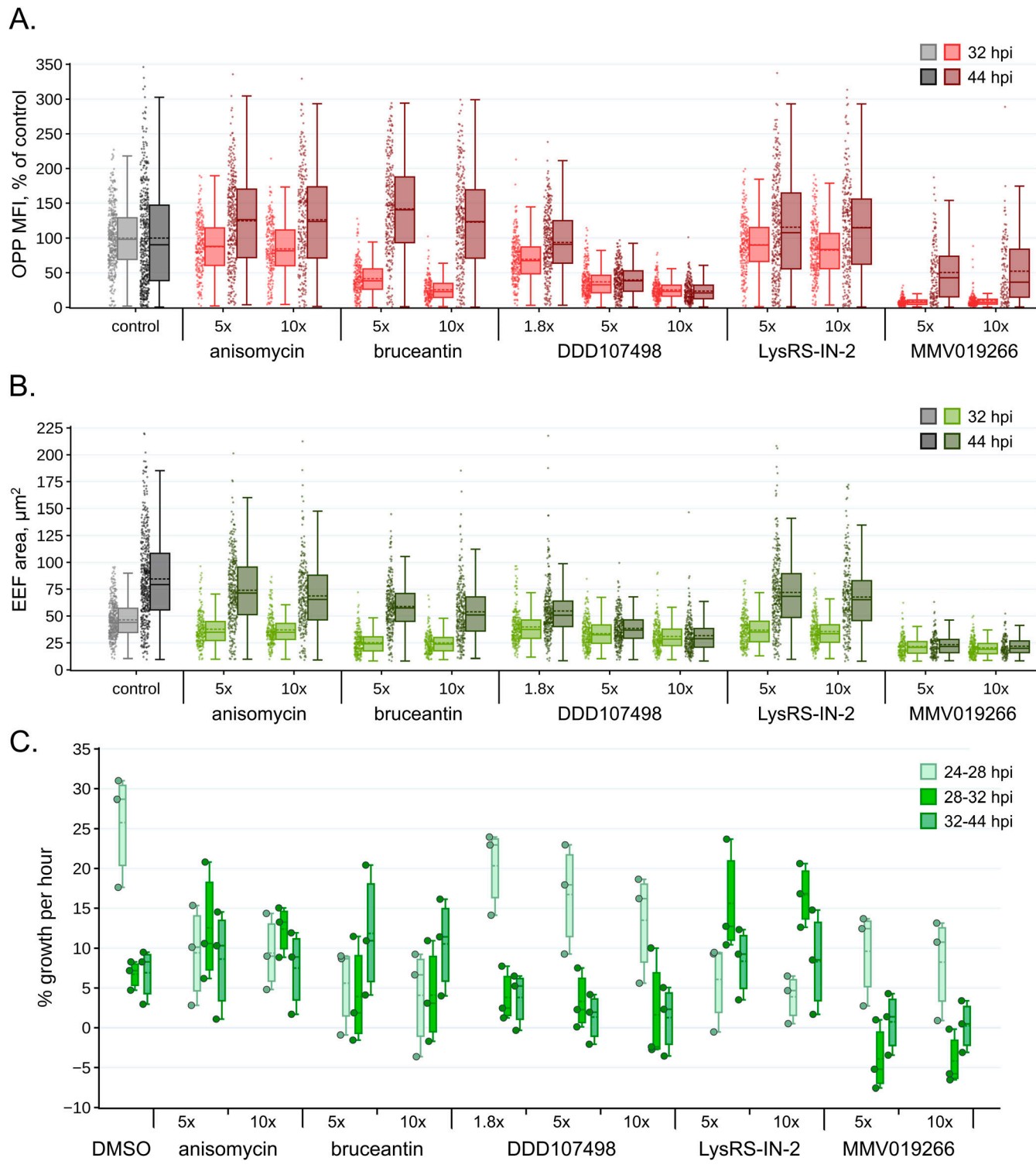

**Figure 3. Post-washout translation recovery and growth of *P. berghei* liver stages.**
**(A, B)** Parasite translation (OPP-MFI, % of control) and (B) parasite area ($\mu m^2$) quantified 4 h (32 hpi) and 16 h (44 hpi) after compound washout, as schematized in Fig 1C; individual points represent single exoerythrocytic forms. **(A, B)** Statistical significance was assessed using an ordinary two-way ANOVA with Tukey's multiple comparisons testing, reported in Table S2. The full dataset can be explored via interactive dashboards in our KNIME hub workflow: https://hub.knime.com/~/spaces/~/~EcnvMwYtqylu2reV/current-state/. **(C)** Modeled % growth per hour during each time window (see the Materials and Methods section for details). Each point represents an experimental mean. Dotted lines in the boxplots represent the mean of n = 3 independent experiments.
Source data are available for this figure.

levels by 44 hpi (Fig 3A and Table S2). MMV019266 treatment also had lasting effects on translation, with little recovery observed 4 h after washout, but on a population level, translation increased to ~50% of DMSO controls after 16 h (Fig 3A). Looking at the individual data points for MMV019266-treated individual parasites at 44 hpi, recovery is quite heterogeneous, with some parasites translating at control levels, whereas others remain completely inhibited (Fig 3A). Translation recovery in the HepG2 cells after anisomycin and bruceantin treatment largely paralleled the LS translation recovery observed (Fig S3B), though bruceantin-treated cells did show greater recovery after 4 h, as would be expected given that bruceantin is somewhat *Plasmodium*-selective.

We next examined parasite growth after compound washout, in terms of parasite size at both recovery timepoints (Fig 3B); from these data, we also estimated % growth per hour for each condition tested, using the assumption that growth rates were linear and constant throughout each time interval for which we had measurements. In control parasite populations, the mean percentage of growth per hour from 24 to 28 hpi was inexpertly more than four times greater than during the 28–32 and 32–44 hpi intervals for all independent experiments, even though growth was variable from one experiment to another (Fig 3C). LysRS-IN-2–treated parasites had the fastest growth rates observed during the 28–32 hpi window, more than double that of the controls (Fig 3C), with anisomycin- and LysRS-IN-2–treated parasites reaching near-equivalent sizes at 32 hpi (Fig 3B and C). Parasites from these two treatment groups continued to grow at a rate greater than or equal to that of the controls from 32 to 44 hpi (Fig 3B and C). This quick and profound growth is well matched to the translation recovery dynamics observed for these compounds (Fig S4). In contrast, the bruceantin-treated parasites were growth-impaired at 32 h, showing only a small increase in mean parasite size (Fig 3B). Just as with translation, bruceantin-treated parasites showed a large increase in size at 44 hpi and the percentage of growth per hour during the 32–44 hpi window was the largest of any treatment group for both 5x and 10x $EC_{50}$ (Figs 3B and C and S4). Despite the robust growth of LysRS-IN-2–, anisomycin-, and bruceantin-treated liver stages at both 5x and 10x $EC_{50}$, parasites from all these groups remained smaller, on average, than controls at 44 hpi (Figs 3B and S4).

DDD107498-treated parasites were strikingly growth-impaired after compound washout. EEFs treated with 5x $EC_{50}$ grew slightly at both 32 and 44 hpi, equivalent to about half the growth rate seen in controls from 28 to 32 hpi, falling to <20% in the 32–44 hpi period (Fig 3B and C), whereas the size of EEFs treated with 10x $EC_{50}$ DDD107498 changed very little from 28 to 44 hpi reflecting profound growth inhibition (Fig 3B and C). Even in the 1.8x $EC_{50}$ treatment, parasites grew slower than the controls, despite the translational recovery seen at both 32 and 44 hpi in the 1.8x $EC_{50}$ DDD107498-treated parasites (Figs 3 and S4). MMV019266-treated parasites showed no growth at all after compound washout (Fig 3B and C). Growth clearly lags behind or is uncoupled from translation for MMV019266-treated EEFs, as the large jump in translational intensity seen in 44 hpi parasites is not at all paralleled (Figs 3 and S4). Our results show that growth does require resumption of translation, as would be expected. Taken together, these results demonstrate that translation inhibition efficacy does not determine either the speed or completeness of LS translation recovery timing nor the extent of *P. berghei* LS parasite growth after washout.

## A brief period of translation inhibition in early schizogony causes both developmental delay and developmental failure in *P. berghei* liver stages

The ultimate readout of LS developmental success in the *P. berghei*–HepG2 infection model is parasite maturation, culminating in the formation of hepatic merozoites and their release from the monolayer inside detached cells. Across all treatment groups, late liver stage development is deleteriously impacted by a brief period of translation inhibition during early schizogony. In the normal course of LS development, the onset of hepatic MSP1 expression in the parasite plasma membrane begins in the late liver stage before cytomere formation and remains through hepatic merozoite formation (see, e.g., reference 32). AMA1 expression begins only as merozoites are being formed (Fig S5), and the release of hepatic merozoite-filled detached cells, the merosome equivalent in the in vitro infection model, into the medium occurs last (30, 33). When parasite populations are analyzed at 72 hpi, there are notable impairments at each of these steps in parasites subjected to translation inhibition during early schizogony, but the level of impairment is compound- and concentration-dependent (Fig 4A). Late liver stage development was most similar in anisomycin (60–63% of control) and LysRS-IN-2 (61–69% of control) treatments (Fig 4A and Table S2). Monolayers treated with bruceantin showed less successful parasite maturation compared to those treated with anisomycin or LysRS-IN-2, particularly at the 10x $EC_{50}$ concentration (Fig 4A). In the 5x and 10x $EC_{50}$ treatments with MMV019266 and DDD107498, very few parasites showed evidence of late liver stage maturation (Fig 4A). The average number of parasites in the MMV019266-treated monolayers was the lowest observed, suggesting that some of the treated parasites may be killed or cleared by the HepG2 cells. In the 1.8x $EC_{50}$ DDD107498 treatments, MSP1-expressing EEFs were reduced to 50% of the controls, and hepatic merozoite abundance was only 17% relative to the controls (Fig 4A). As a whole, these results were again not compatible with translation inhibition efficacy determining downstream antiplasmodial effects.

The harmful effects of a brief period of translation inhibition in early schizogony are even more apparent in 72 hpi merosome counts, with no merosomes collected from any of the DDD107498- or MMV019266-treated wells, and very few in the 10x $EC_{50}$ bruceantin wells (Fig 4A). We anticipated that a 4-h period of translation inhibition would cause a small developmental delay, and thus our matched assessment of monolayer and detached cells/merosomes at 72 hpi, but we also wanted to be able to separate more profound cytostatic effects from those that likely indicate developmental failure and compound cytotoxicity. Thus, we set up additional samples for repeated merosome harvesting from a single well, with collections made at 72, 96, and 120 hpi (see Table S3 for merosome counts per experiment). Control parasites that completed hepatic merozoite formation were largely released from the monolayer by 72 hpi, with no control merosome formation during the 96–120 hpi period (Fig 4B). Looking at total merosome counts over 3 d, every condition except 10x $EC_{50}$ DDD107498 ultimately produced merosomes, with both LysRS-IN-2–treated samples producing the most (Fig 4B). Strikingly, the 1.8x $EC_{50}$ DDD107498-treated parasites were nearly equivalent to 5x and 10x $EC_{50}$ LysRS-IN-2 in overall merosome

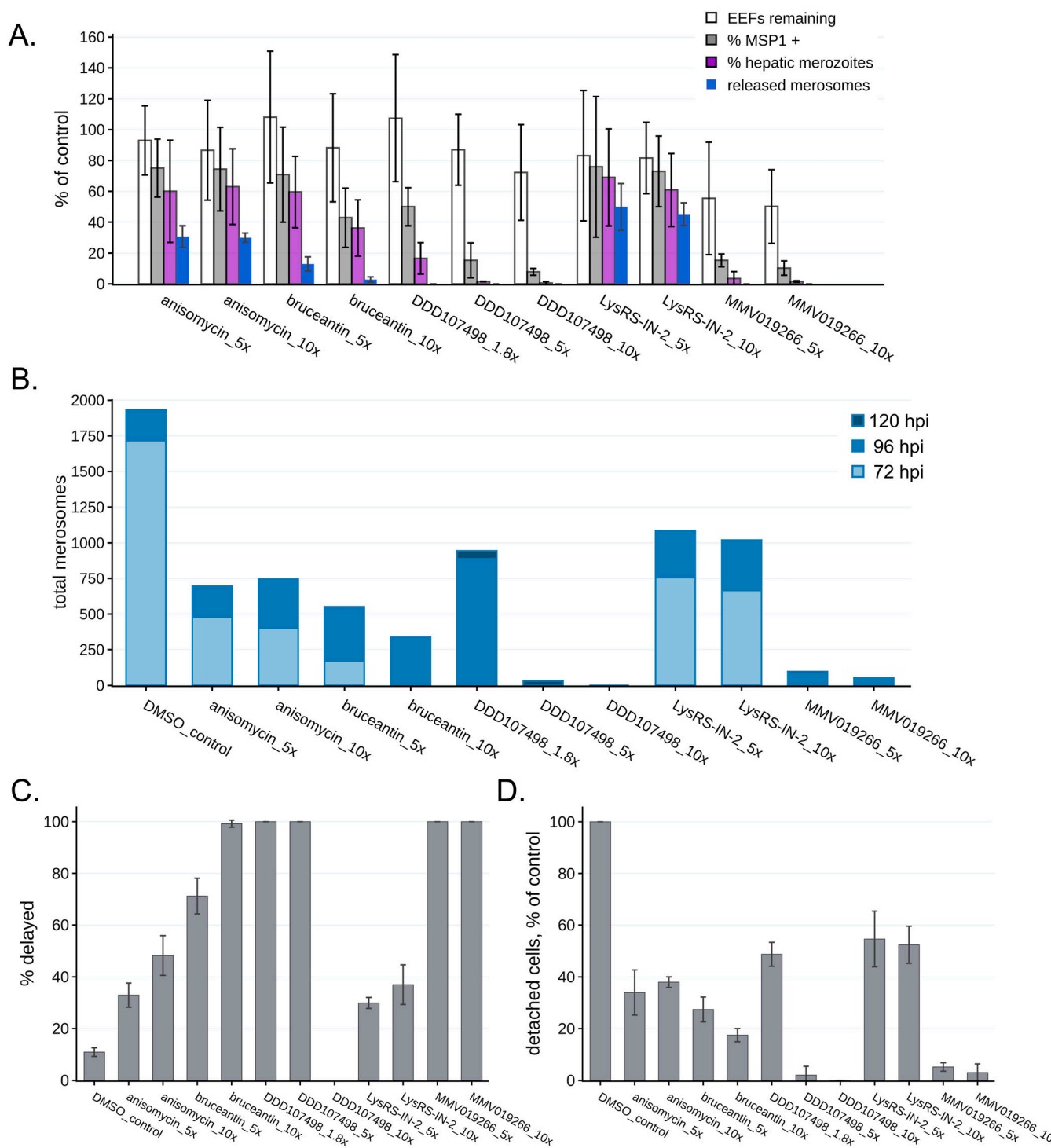

**Figure 4. Translation inhibitor effects on late liver stage development.**
**(A)** Parasite maturation at 72 hpi, normalized to DMSO controls set at 100%, is quantified based on exoerythrocytic forms (EEFs) remaining in the monolayer, the percentage of monolayer EEFs expressing merozoite surface protein 1, the percentage of monolayer EEFs containing hepatic merozoites (expressing both merozoite surface protein 1 and apical membrane antigen 1; see Fig S5), and merosomes released. Each treatment well contained a coverslip, which was used for monolayer maturation analysis, and the medium contained detached merosomes, which were collected and counted. Bars report the mean of n = 3 independent experiments, error bars = SD. **(B)** Detached merosomes were collected and counted in experiment-matched wells at 72 (n = 3 expts), 96 (n = 3 expts), and 120 hpi (n = 2 expts), with stacked bar charts showing the total merosomes per timepoint. **(C)** Percentage of total detached cells/merosomes released after 72 hpi (% delayed) are reported as means with error bars showing the SD. **(D)** Total detached cell/merosome release normalized to the DMSO controls, with error bars showing the SD. For all panels, treatments are labeled by compound and multiple of EC$_{50}$.

production, but the merosomes were only found in the 96 and 120 hpi collections (Fig 4B). Separating the data into cytostatic (% of delayed merosomes released after 72 hpi, Fig 4C) and potentially cytotoxic (% of total control merosomes, Fig 4D) effects, we find that all treatments that led to any merosome production caused both cytostatic and cytotoxic effects on parasite populations. A clear difference can be seen between 5x and 10x $EC_{50}$ anisomycin treatments, with the latter causing more cytostatic effects (Fig 4C), even though the eventual merosome production between the two was equivalent (Fig 4D). Despite inducing essentially total translation inhibition at both 5x and 10x $EC_{50}$ (Fig 2A), bruceantin was again the compound with the most variability between the 5x and 10x $EC_{50}$ treatments in terms of antiplasmodial effects, as cytostatic effects were lower and overall developmental success was higher at the 5x $EC_{50}$ concentration (Fig 4B–D). Brief treatment during early LS schizogony with MMV0192667 at both 5x and 10x $EC_{50}$ had extremely strong antiplasmodial effects, with overall developmental success of ~5% and 3%, respectively (Fig 4B and D), and though all merosomes were delayed, they formed in all three experiments (Table S3). Brief DDD107498 treatment at 5x and 10x $EC_{50}$ also appeared to overwhelmingly kill EEFs, with merosomes produced only after 96 hpi in a single experiment, and only in the 5x $EC_{50}$ treatment (Fig 4B and D and Table S3).

### Correlations between translation inhibition efficacy and later *P. berghei* liver stage antiplasmodial effects are compound-specific

Even with our limited compound set, it is clear that translation inhibition efficacy does not determine the speed or completeness of parasite recovery after a brief treatment with translation inhibitors. After compound washout, parasite translation, growth, and development appeared quite variable between compounds. To determine how well-correlated individual metrics were, both overall and for individual compounds, we performed a correlation analysis of several key metrics describing the parasite response to acute translation inhibition. First, we examined correlations using data from all five compounds at both 5x and 10x $EC_{50}$. The only strong correlation seen across data from all compound treatments was a negative correlation between growth recovery from 28 to 32 hpi and growth recovery from 28 to 44 hpi (Fig 5A; see Table S3 for statistics). Analyzing the compounds individually, strong correlations that are compound-specific are seen, with clear differences even for anisomycin and LysRS-IN-2 treatments, which were the two most similar compounds analyzed. For instance, in the LysRS-IN-2 treatments, there are strong positive correlations between translation recovery from 28 to 44 hpi and the MSP1 expression (correlation = 0.82, *P* = 0.001) and hepatic merozoites (correlation = 0.73, *P* = 0.007) (Table S3), whereas in the anisomycin treatments, they are not correlated (translation recovery versus MSP1, correlation = 0.03; translation recovery versus hepatic merozoites, correlation = 0.10) (Table S3). In addition, there is a weak, yet significant, correlation between translation inhibition efficacy and growth inhibition at 28 hpi (correlation = 0.65, *P* = 0.02) in LysRS-IN-2 treatments, but not in anisomycin (correlation = –0.07). This difference may be supported by the observation that on average, parasites grew more during the anisomycin treatments compared with the LysRS-IN-2 treatments (Fig 3C and Table S3), even though anisomycin was a more efficacious translation inhibitor (Fig 3A).

When a principal components analysis is applied to the same set of features, but with data separated by both concentration and experiment, most compounds are clearly separated from one another, whereas anisomycin and LysRS-IN-2 are partially overlapping (Fig 5B). The 5x and 10x $EC_{50}$ treatments of MMV019266 were essentially indistinguishable and also had the greatest similarity between experiments of the compounds tested (Fig 5B). Overlap is also seen for the two anisomycin and LysRS-IN-2 concentrations, but not for 5x and 10x $EC_{50}$ bruceantin or DDD107498 treatments (Fig 5B). Unsurprisingly, the 1.8x $EC_{50}$ DDD107498 data points are distant from the other DDD107498 concentrations (Fig 5B).

### DDD107498 exerts antiplasmodial effects on translationally arrested parasites

Though DDD107498 at both 5x and 10x $EC_{50}$ concentrations was the least efficacious translation inhibitor tested, it had the strongest antiplasmodial effects. Strikingly, the 1.8x $EC_{50}$ concentration (20 nM) caused only ~60% reduction in parasite translation output and the least growth inhibition during the treatment period, yet had stronger effects on translation recovery and growth after washout (Figs 3 and S4) and on LS developmental success (Fig 4), than 5x $EC_{50}$ concentrations of anisomycin and Lys-RS-IN-2. Given that in vitro antiplasmodial potency of DDD107498 is consistently between 1 and 2 nM (Table S1 and (5)), we tested whether DDD107498 achieves a more profound LS translation inhibition after prolonged treatment. Treatment with 20 nM DDD107498 for 24 h, from 24 to 48 hpi, did not lead to increased LS translation inhibition (Fig S6). Taken together, these data led us to wonder whether DDD107498 can exert antiplasmodial effects independent of its ability to inhibit translation. To test this, we designed an experiment to determine whether DDD107498 can exert antiplasmodial effects on translationally arrested *P. berghei* LS parasites. A supra-maximal concentration of anisomycin (10 $\mu$M, equivalent to 38x the translation inhibition $EC_{50}$) was added to *P. berghei* LS at 24 hpi to shut down translation, followed by the addition of 20 nM DDD107498 at 25 hpi, thorough compound washout at 28 hpi, and analysis 20 h later at 48 hpi (Fig 6A). We have previously demonstrated that LS parasites recover from this anisomycin treatment regimen in terms of both translation and growth and that 10 $\mu$M anisomycin treatment for 30 min is sufficient to cause essentially complete inhibition of *P. berghei* LS translation (17). Strikingly, anisomycin-arrested parasites subsequently treated with DDD107498 were smaller on average and had lower mean translational output than either anisomycin or DDD107498 single-treatment controls (Fig 6B and C). We next used the same treatment regimen, but read out antiplasmodial efficacy with merosome collections at 72, 96, and 120 hpi (Fig 6D). Anisomycin-arrested EEFs subsequently treated with DDD107498 had a marked decrease in merosome formation with respect to the DMSO, anisomycin, and DDD107498 controls (Figs 6E–G and S7A, Table S4). A 3-h DDD107498 treatment gave a highly similar merosome production profile (Fig 6E) compared with the 4-h treatment (Fig 4B). DDD107498-treated anisomycin-arrested parasites showed a near-identical cytostatic effect as those treated with DDD107498 alone, and completely distinct from those treated only with anisomycin (Fig 6F). Essentially, all the difference between DDD107498 treatment of translating versus anisomycin-arrested parasites was

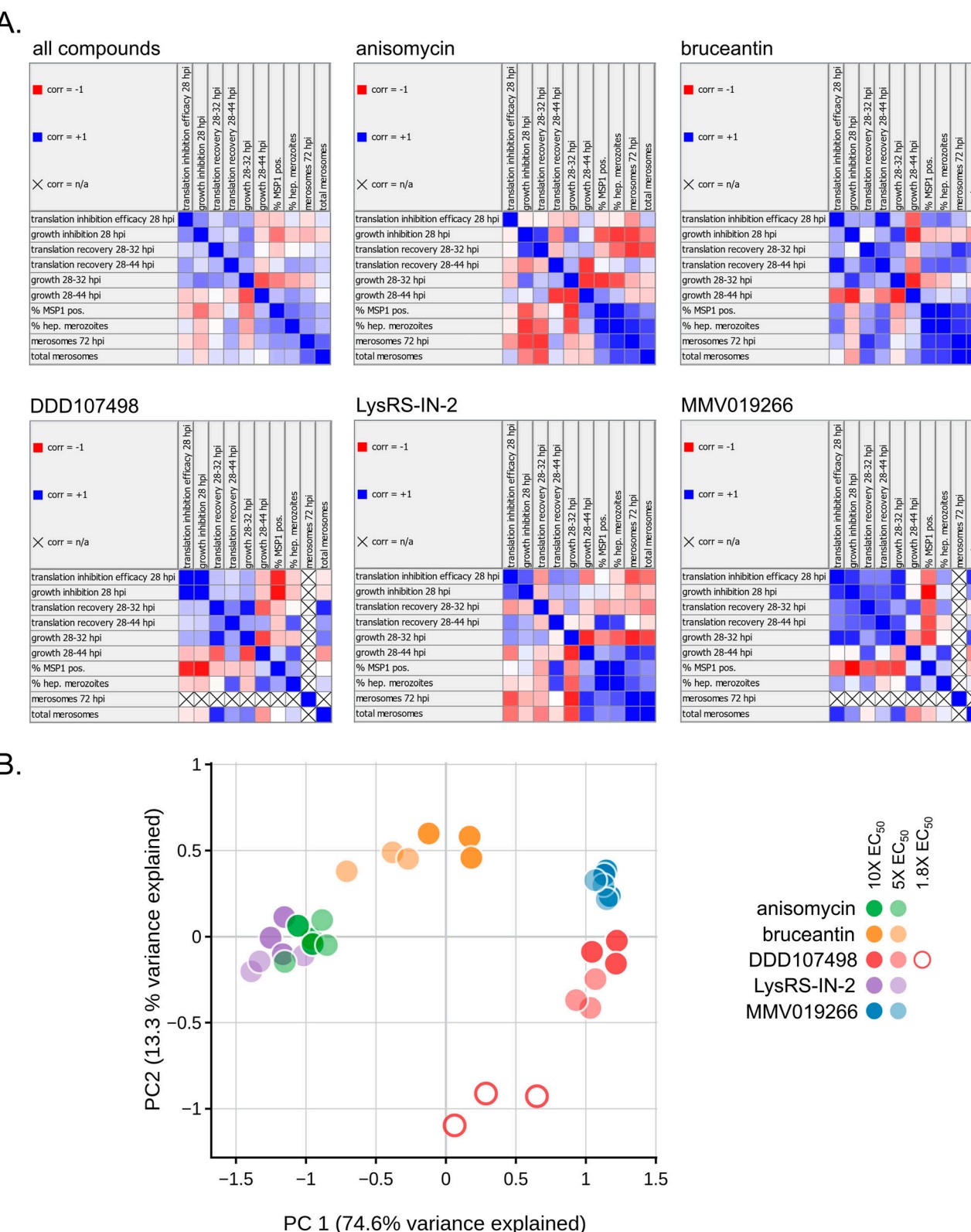

**Figure 5. Multivariate analysis of translation inhibitor efficacy, recovery, and antiplasmodial effects on *P. berghei* liver stage parasites.**
**(A)** Linear correlation matrices comparing compound efficacy, parasite recovery, and development (Pearson's correlation, two-sided; correlation values and *P*-values reported in Table S2). Metrics used in correlation analysis were experimental means reported in Table S2, or calculated from them, with 5x and 10x EC$_{50}$ concentrations combined (see the Materials and Methods section for details). Positions are marked X when one of the two metrics being compared has a value of 0. **(A, B)** Principal

reflected in total merosome production up to 120 hpi (Fig 6G), reflecting a likely increase in parasite death. Given these unambiguous results, we repeated these experiments using 5x translation inhibition $EC_{50}$ concentrations of the three other translation inhibitors studied (Table S1) to shut down parasite protein synthesis before the addition of DDD107498, while maximizing the difference in antiplasmodial profile of each compound alone versus that of DDD107498. In each case, the addition of DDD107498 to the translationally arrested parasites led to 100% delayed merosome release, with LysRS-IN-2– and bruceantin-treated parasites shifted to the phenotype of DDD107498-treated parasites (Figs 6H and I and S7B, Table S4). The addition of DDD107498 to translationally arrested parasites exerted a remarkably clear and stable increase in likely parasite killing, with at least 50% reduction in total merosomes formed, regardless of whether anisomycin, bruceantin, LysRS-IN-2, or MMV019266 was used to inhibit parasite protein synthesis (Fig 6H and J).

## Discussion

Our systematic assessment of the relationship between the translation inhibition efficacy and liver stage antiplasmodial efficacy of known translation inhibitors generated several intriguing results worthy of follow-up study, and the firm conclusion that the extent of translation inhibition such a compound induces does not determine the strength of its downstream antiplasmodial effects. This is most dramatically demonstrated by the least efficacious translation inhibitor tested, DDD107498, which had the strongest antiplasmodial effects at both 5x and 10x translation inhibition $EC_{50}$ concentrations. This unexpected disconnect between translation inhibition efficacy and antiplasmodial efficacy is evident throughout our data, clearly demonstrating that translation inhibition efficacy alone is not a suitable biomarker for antiplasmodial efficacy of translation inhibitors. Correlation and principal components analyses indicated that each compound displays a specific antiplasmodial effect profile downstream of the translation inhibition it induced, with no single metric strongly correlated to merosome production across all the compounds. The mechanism also does not appear likely to be informative, though our test set was quite small. LysRS-IN-2 and anisomycin were by far the two most similar compounds across all metrics analyzed, despite having no mechanistic similarity whatsoever upstream of their effects on protein synthesis. Anisomycin inhibits translation elongation (34), and structural data in other eukaryotes indicate that it binds to the A-site of the 60S ribosome subunit (22), whereas LysRS-IN-2 inhibits the *P. falciparum* lysyl-tRNA synthetase (Pf cKRS), which catalyzes aminoacylation of lysine onto its cognate tRNA for incorporation into nascent polypeptides (28). Taken together, our data suggest that the no single metric can provide an adequate correlate of LS antiplasmodial effects of translation inhibitors; instead, quantification of total merosome production up to 120 hpi is needed. Quantifying merosome production unfortunately relies on manual counting, making it both time-consuming and error-prone. Reproducible and automated assays that allow for merosome, or better yet individual hepatic merozoite, quantification will be needed.

Among our surprising findings in this work was the strength of antiplasmodial effects exerted by a 4-h compound treatment during early LS schizogony. We chose the 4-h treatment based on previous findings (17): (1) these compounds all achieve profound translation inhibition within 30 min, suggesting that a 4-h treatment ensures several hours of maximal inhibition, and (2) high-concentration anisomycin treatment for 4 h was reversible, in terms of both translation and parasite growth, after washout. We did not anticipate that the weakest effect we would observe on parasite developmental success after compound washout would be ~50% reduction in the release of hepatic merozoite-filled merosomes/detached cells. This indicates substantial heterogeneity in individual parasites' abilities to withstand drug pressure, and a deeper understanding of which individuals are killed by the most reversible compounds may yield valuable insights.

Parasite killing speed is a key consideration for antimalarial therapeutic use, as rapid reductions in parasite burden can directly influence clinical outcomes. Speed-of-kill in the *P. falciparum* ABS is determined using PRR assays (27, 35), in which mixed-stage parasite cultures are subjected to compound treatment for 120 h, with aliquots drawn every 24 h that are thoroughly washed, and then put back into compound-free culture with limiting dilution, and monitored for several weeks to detect parasite recrudescence and calculate the number of viable parasites present. When compounds are compared at equivalent effective concentrations, PRR assays are able to reveal profound compound-specific differences that are masked in standard 48- or 72-h growth assays, where compounds often appear equally effective (36). Similarly, in standard liver stage assays that quantify EEF biomass (37, 38), typically after 48 h of continuous compound treatment, all compounds active against the parasite at the earliest stage of development will present a similarly dramatic reduction in biomass. Parasites treated with saturating concentrations of DDD107498, LysRS-IN-2, or MMV019266 are indistinguishable in our 48 hpi live luciferase assay, with essentially complete biomass reduction (ranging from 0.7 to 1.5% of the DMSO control). As the LS does not cycle, a PRR-like assay is challenging to implement. In the absence of an integrated liver-to-blood infection model, LS viability must be assessed in the treated population. Yet, the remarkable phenotypic separation we see between treatment groups indicates that this assay modality could yield useful insights across compound classes into liver stage speed-of-kill (or perhaps speed-of-effect), with the clear caveat that we cannot be certain that parasites which fail to release merosomes by 120 hpi are truly non-viable. Because of the likelihood of stage-specific compound activity in the LS, it might be worthwhile to run parallel assays targeting, for example, 4–8 hpi and 48–52 hpi, to capture differences in early and late LS activity. For translation inhibitors, at least, the early schizogony treatment workflow is quite informative. Notably, the two aaRS inhibitors, LysRS-IN-2 and MMV019266, display profound differences in speed-

components analysis using the same metrics analyzed in (A), but with data separated by both concentration and individual experiments. The first two principal components explain a total of 87.9% of the variance.

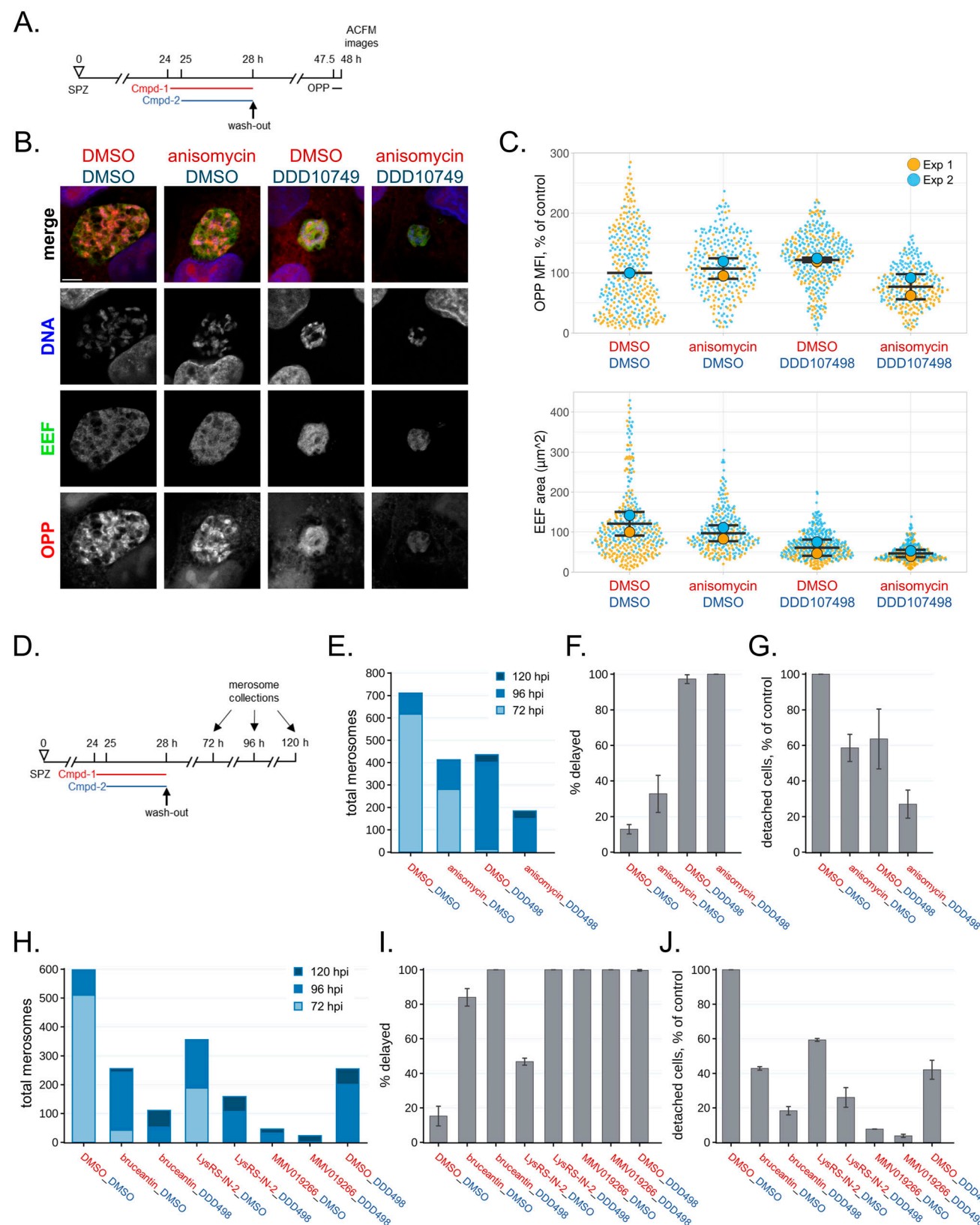

**Figure 6. Antiplasmodial effects of DDD107498 on translationally arrested *P. berghei* LS parasites.**
**(A, B, C)** Experimental schematic for panels (B, C) quantifying translation recovery and growth at 48 hpi after DDD107498 treatment of anisomycin-arrested parasites in early schizogony. **(B)** Representative single confocal images of translation in *P. berghei* LS at 48 hpi. Merged images are pseudocolored as indicated with parasite

of-effect/kill, similar to what is observed in PRR assays. LysRS-IN-2 is slower to kill mixed *P. falciparum* ABS cultures than even atovaquone, the slowest-acting reference compound, whereas MMV019266 and similar thienopyrimidines kill as quickly as artemisinin, the fastest-acting reference compound (26, 28). LysRS-IN-2 was the most reversible compound we tested, and yet still able to reduce merosome release by ~50% at both 5x and 10x translation inhibition $EC_{50}$, whereas MMV019266 reduced developmental success to ~5% at both 5x and 10x $EC_{50}$, with substantial developmental delay. In contrast, DDD107498 tested at 5x and 10x translation inhibition $EC_{50}$ was the most efficacious antiplasmodial compound that we tested, with not a single merosome recorded at the 10x $EC_{50}$ concentration in any experiment. This does not parallel its *P. falciparum* PRR profile, which is slow-killing, like atovaquone (5).

DDD107498 presents a rather puzzling case as a translation inhibitor. We detected no LS translation inhibition activity from 1 nM DDD107498 in our earlier study, and only ~40% inhibition at a 10 nM concentration (17), roughly the lower end of a 10x LS biomass $EC_{50}$ estimate. This disconnect is neither liver stage nor *P. berghei*-specific. Using a *P. falciparum* ABS lysate–based approach, the translation inhibition $IC_{50}$ for DDD107498 was determined to be 60.5 nM (19), whereas testing at 10x the antiplasmodial $EC_{50}$ led to ~45% or ~70% translation inhibition using $^{35}$S-labeled amino acid incorporation assays in *P. falciparum* ABS populations (5, 19). Because of the nearly 10-fold increase in translation inhibition $EC_{50}$ versus LS biomass inhibition $EC_{50}$ for DDD107498, and its place among promising new antimalarial compounds progressing in clinical trials (6, 7), we also ran DDD107498 at 20 nM—a rough, upper estimate of 10x $EC_{50}$ for LS biomass inhibition, as this concentration supports more direct comparison with other data on the compound's antiplasmodial efficacy. We also note that previous work using a *P. berghei*–HepG2 spheroid model previously found that a 25 nM concentration was sufficient to completely inhibit parasite growth during the 24–48 hpi treatment window (39). The 20 nM concentration (1.8x LS translation inhibition $EC_{50}$) induced ~60–65% translation inhibition, consistent with the shallow slope of the fit concentration–response curve. Despite this, the 1.8x $EC_{50}$ DDD107498 treatment induced stronger antiplasmodial effects than anisomycin and LysRS-IN-2 at both 5x and 10x $EC_{50}$. The 1.8x DDD107498-induced phenotypes were so different from those of 5x and 10x $EC_{50}$ DDD107498, that it appears to be essentially a distinct compound in the PCA. Given these results, we designed an experiment to ask whether 20 nM DDD107498 could exert antiplasmodial effects on EEFs translationally arrested by a supersaturating concentration of anisomycin for an hour before DDD107498 addition. Adding DDD107498 to these translationally arrested parasites (~99% translation inhibition expected, per the 10x $EC_{50}$ data) not only induced the same cytostatic effects caused by DDD107498 alone, but the staggered combination treatment further reduced merosome formation by ~50% compared with anisomycin

alone. DDD107498 induced a similar ~50% reduction in developmental success in parasites translationally arrested by the other three compounds in our study. We note that these experiments were not designed to test for synergy between compounds, and are insufficient to do so, but we interpret the data as more suggestive of additive effects. It will be worthwhile to look at this more carefully, as the combination of multiple active compounds into a single antimalarial medicine is critical for preventing blood stage resistance from developing (11), but may also effect superior chemoprotection against the LS. The unexpected synergy between bacterial translation inhibitors has been documented (40), and it seems worthwhile to test for such synergy between antiplasmodial translation inhibitors in the LS, though the merosome release readout is problematic to scale.

Baquero and Levin (41) provide a useful framework emphasizing the distinction between proximate versus ultimate causes of antibacterial action by antibiotics, and the antiplasmodial activity of DDD107498 can be considered in this light. Is the proximate cause of DDD107498 LS antiplasmodial activity mediated by a target other than eEF2? This seems quite unlikely, given the centrality of eEF2 to protein synthesis and the overwhelming similarity of DDD107498 potency across species and stages (5), but direct examination of LS parasites with DDD107498-resistant eEF2 alleles will ultimately be needed. The genetic evidence linking eEF2 mutations to DDD107498 resistance is extremely strong, and no other parasite genes have been implicated to date (5, 6, 20, 21); furthermore, a mutation that shifts *P. falciparum* ABS antiplasmodial potency also shifts *P. falciparum* ABS translation inhibition potency (5). It is notable that eEF2 has roles beyond catalysis of the translocation step of elongation, which are also potential proximate causes of DDD107498 activity. eEF2 is involved in the fidelity of ribosomal translocation via its diphthamide modification (42, 43) and catalyzes reverse translocation (44). In the presence of a compound able to stabilize eEF2 on the ribosome, as has been demonstrated for sordarin (45), the likelihood of reverse movement is increased, which has been suggested to contribute to sordarin's activity (44). Though we detect substantial translation in the presence of 20 nm DDD107498, the makeup of the nascent proteome under these conditions is unknown. If translation fidelity or processivity is affected, the polypeptides synthesized might be non-functional and sufficiently aberrant to induce cellular stress, a possible ultimate cause of antiplasmodial activity that is largely consistent with our observations. However, the idea that DDD107498 activity could be ultimately due to a toxic nascent proteome is hard to reconcile with the fact that DDD107498 exerted antiplasmodial activity against parasites that were translationally arrested during the entirety of their exposure to the compound. This could possibly be explained if DDD107498 was retained in the parasite after compound washout, perhaps in a stable complex with binding partners. DDD107498 could also exert LS antiplasmodial effects on processes in addition

(exoerythrocytic form) immunolabeled with *α*-HSP70, OPP-A555 labeling the nascent proteome, and DNA stained with Hoechst. Scale bar = 5 *μm*. **(C)** Single parasite translation and size quantified at 48 hpi. Single points show individual exoerythrocytic forms color-coded by independent experiments, with experimental means represented by large circles (n = 2), and bars represent the mean and SD of the experiments. **(D)** Experimental schematic for quantifying merosome release in experiment-matched wells at 72, 96, and 120 hpi, after DDD107498 treatment of translationally arrested early LS schizonts and subsequent washout. Stacked bar charts **(E, H)** report the total merosomes collected per timepoint from all experiments. **(F, I)** Percentage of total detached cells/merosomes released after 72 hpi (% delayed) are reported as means with error bars showing the SD. **(E, F, G, H, I, J)** Total detached cell/merosome release normalized to the DMSO controls, with error bars showing the SD. Data are shown from n = 3 (E, F, G) or n = 2 (H, I, J) independent experiments; DDD498 = DDD107498 in the axis labels for (E, F, G, H, I, J).

to protein synthesis, via hypothetical non-canonical roles of eEF2 (as described for other elongation factors ([46], [47])) or via additional targets. As we show that prolonged incubation with 20 nM DDD107498 does not lead to increased translation inhibition in *P. berghei* EEFs, the hypothesis that translation inhibition becomes stronger with prolonged treatment can be rejected.

Antibacterial translation inhibitors have been extensively studied ([48], [49]), yet far less is known about their ultimate killing mechanisms. It has been suggested that bactericidal antibiotics, including those that target the ribosome, ultimately kill bacteria by stimulating the production of hydroxyl radicals, whereas bacteriostatic antibiotics, regardless of mechanism, do not ([50]), though this has been controversial ([51]). Antibiotics targeting protein synthesis fall into both bactericidal and bacteriostatic categories, sometimes even within the same class of molecules, such as the macrolides, for which evidence exists that cidality is linked to slow dissociation rates from the ribosome ([52]). For aminoglycosides, clinical mainstays that inhibit protein synthesis by binding with high affinity to the ribosomal A-site ([53]), several hypotheses for the ultimate cause of bacterial killing have been proposed, including ribosome-independent but not alternative direct effects on the bacterial outer membrane (reviewed in reference [41]). Comparable compound-specific systemic effects leading to parasite killing could be at play for antimalarial translation inhibitors, as far less is known about their ultimate parasite killing mechanisms. Our data as a whole suggest these ultimate killing mechanisms are worthy of investigation, as the hypothesis that *P. berghei* LS antiplasmodial killing by translation inhibitors is determined by translation inhibition efficacy per se should be rejected.

The rate of protein synthesis directly influences the rate of cellular growth, which is in addition modulated by cell-extrinsic factors such as nutrient availability, along with many intrinsic factors, such as protein concentration, ribosomal capacity, and cell size ([54], [55], [56], [57]). In organisms where replication culminates in binary fission, including several bacterial, fungal, and metazoan species, cellular growth rates have been observed to be exponential ([54], [56], [58]), where absolute growth is directly proportional to cell size, though this remains somewhat controversial ([59]). We used an assumption of constant, linear growth to estimate % growth per hour for *P. berghei* EEFs based on data collected at 24, 28, 32, and 44 hpi, which led to the unexpected finding that % growth per hour was ~4 times greater during the 24–28 hpi period, than for the other two. Maintaining an appropriate nuclear-to-cytoplasmic ratio is crucial for growth and cellular functions ([60], [61]); because both DNA content and cell size increase immensely during *Plasmodium* EEF growth, there should be no reason to assume that growth must slow as the parasite gets larger. Although our data are sparse, it does suggest that growth is not constant, and that the *P. berghei* LS cellular state may be front-loaded to achieve rapid growth earlier in development. Despite the variability in growth rates and translational intensity seen between independent experiments, the growth rate variability between timepoints was robustly observed in each experiment. Although we were not able to identify a technical explanation for the differences in growth rates, particularly because some compound-treated samples exhibited high growth rates after washout in both the 28–32 and 32–44 hpi windows in which control growth was slow, experiments purposely

designed to query LS parasite growth rates over time will be needed to validate our preliminary findings.

Because of the high levels of translation inhibition efficacy we observed at 28 hpi (>90% for all compounds except DDD107498) and evidence that most compounds reach similar levels of inhibition after only 30 min of treatment ([17]), it is notable that parasite growth inhibition was not more profound during the 4 h in which the compound was present; none of the compounds reduced growth from 24 to 28 hpi by even 50%. This could suggest that a small amount of residual translation is enough to support some parasite growth. Alternatively, parasite growth may be driven by a high steady-state protein concentration at the onset of treatment. In fission yeast, cellular growth appears to function as a mechanism controlling intracellular protein concentrations ([56]). When yeast cells were prevented from increasing in size while protein synthesis continued at a rate equivalent to log growth, thus artificially increasing the concentration of protein in the cells, the yeast entered a phase of hypergrowth when released from the growth constraints. Interestingly, LysRS-IN-2 induced what could be interpreted as a hypergrowth phenotype immediately after compound washout; the percent growth per hour of LysRS-IN-2–treated parasites during the 28–32 hpi window was nearly as high as the growth exhibited by the controls during the 24–28 hpi period. Bruceantin-treated parasites remained profoundly growth-inhibited for the first 4 h after washout despite the partial recovery of translation during the same time period, but had the fastest growth rate during the 32–44 hpi window. This suggests that the cellular state facilitating rapid growth in control parasites at 24 hpi remains unchanged in parasites that are translationally arrested at that time, and this cellular state could be used for a rapid growth phase after compound removal once translation levels have sufficiently recovered. Further work will be needed to assess how generalizable these results are across different treatment windows during the *P. berghei* LS and those of the clinically relevant *Plasmodium* spp. Considering the magnitude of growth preceding the cytokinesis event that forms hepatic merozoites, the *Plasmodium* liver stage parasite will be a fascinating system for further studies of growth mechanisms and their integration with translational output.

# Materials and Methods

### HepG2 cell culture and infection by *P. berghei* sporozoites

Human hepatoma (HepG2) cells were cultured in complete DMEM (10313-021; Gibco) supplemented with 10% (vol/vol) FBS, 1% (vol/vol) GlutaMAX (35050-061; Gibco), and 1% (vol/vol) penicillin–streptomycin (15140-122; Gibco), and maintained at 37°C, 5% $CO_2$. Sporozoites were isolated from *P. berghei*-infected *Anopheles stephensi* mosquitos (New York University Insectary and University of Georgia SporoCore); sporozoite isolation and infection of HepG2 cells were performed as previously described ([17]). Briefly, *P. berghei* sporozoites (ANKA 676m1cl1 from BEI Resources, MRA-868) with dual reporter expression construct (EEF1a promoter–controlled firefly luciferase and GFP fusion protein ([62])) were extracted from *A. stephensi* mosquito salivary glands, counted, and diluted

into complete DMEM further supplemented with 1% (vol/vol) penicillin–streptomycin–neomycin (15640-055; Gibco), 0.835 $\mu$g/ml amphotericin B (15290-018; Gibco), 500 $\mu$g/ml kanamycin (30-006-CF; Corning), and 50 $\mu$g/ml gentamycin (15750-060; Gibco) (iDMEM), before being exposed to HepG2 monolayers, centrifuged at 2,000$g$ for 5 min, and incubated, as described, for 2 h. After incubation, monolayers were washed with PBS and replenished with iDMEM for direct infections occurring in 24-well plates with or without glass coverslips. For experiments carried out in 96-well plates (655098; Greiner), infected cells were detached using TrypLE Express (12605-028; Gibco), washed, counted, and reseeded at the desired density. All *P. berghei*-infected HepG2 cells were maintained at 37°C, 5% $CO_2$. Live luciferase activity after 48-h treatment was performed as previously described (25).

**OPP labeling, compound treatments, and washouts**

20 $\mu$M OPP (Invitrogen C10459) was used to label cells for 30 min at 37°C, before 15-min fixation with PFA (30525-89-4; Alfa Aesar) diluted to 4% in PBS at RT (please see reference 63 for a detailed protocol). Anisomycin (176880; EMD Millipore/Sigma-Aldrich), bruceantin (HY-N0840; MedChem Express), DDD107498 (A8711; Apex Bio), Lys-RS-IN-2 (Y-126130; MedChem Express), and MMV019266 (STK845176; Vitas-M Laboratory) were solubilized in DMSO (D2650; Sigma-Aldrich), aliquoted, and stored at –20°C. Acute pre-treatment assays were performed as previously described (17). Briefly, compounds were applied for 3.5 h followed by 30-min OPP labeling in the presence of compound treatment. For the competitive OPP (co-OPP) assay performed in Fig S1, LysRS-IN-2 and OPP were applied simultaneously for 30 min. Equimolar concentrations of DMSO (0.1–0.2% vol/vol) were maintained for all DMSO controls and treatments. For samples to be analyzed after 28 hpi, samples were subject to a stringent compound washout protocol, where the wash volume and total number of media exchanges were calculated to ensure a ≥4-log reduction in compound concentration per well. At the end of the 4-h treatment, fresh iDMEM was added to each well to triple the final volume, reducing 10x $EC_{50}$ concentrations to 3.3x $EC_{50}$. Next, all media were removed by pipette and replaced with fresh iDMEM, for a total volume of 1200 $\mu$l per well in 24 wp, and 300 $\mu$l per well in 96 wp. This wash step was repeated for a total of three full washes. Based on the average retained iDMEM volume in each well type (80 $\mu$l in 24 wp and 20 $\mu$l in 96 wp), the 10x $EC_{50}$ treatment should be exposed to ≤0.001x $EC_{50}$ after the last wash.

**Click chemistry and immunofluorescence**

Click chemistry was performed using copper (I)–catalyzed cyclo-addition of Alexa Fluor 555 picolyl azide using Invitrogen Click-iT Plus AF555 (C10642; Invitrogen), largely according to the manufacturer's protocol. All click reactions were performed using a 1:4 $Cu_2SO_4$ to copper protectant ratio. *P. berghei*-infected HepG2 cells were immunolabeled using anti-PbHSP70 (1:200; 2E6 mouse mAb) (64) to mark EEFs followed by Alexa Fluor 488–labeled donkey anti-mouse IgG (A21202; Invitrogen). MSP1 was immunolabeled with Anti-PBANKA_083100, affinity-purified rabbit polyclonal (peptide SSTEPASTGTPSSGC, produced by GenScript, 1:1,000), followed by Alexa Fluor 555–labeled donkey anti-rabbit IgG (A31572; Invitrogen),

and AMA1 was labeled using *P. falciparum* rat monoclonal 28G2 (65) (MRA-897, 1:300; BEI Resources), followed by Alexa Fluor Plus 647–labeled donkey anti-rat IgG (A48272; Invitrogen). All secondary antibodies were used at a concentration of 1:500. DNA was stained with Hoechst 33342 (62249; Thermo Fisher Scientific) (1:1,000). All labeling was carried out in 2% BSA in PBS.

**Image acquisition**

ACFM imaging was performed using a Leica SP8 confocal microscope using an HC PL APO 63x/1.40 oil objective for glass coverslips (Figs 6B and C, S1A, B, and D, and S5), or an HC PL APO 63x/1.40 water objective for 96-well $\mu$clear plates. Images in Fig S1D were acquired manually and were processed using Fiji (66). For Fig 4A, HCI images were acquired using a 10x air objective. All other images presented in this study, as well as images used for quantitative analysis, were acquired using ACFM as previously described (17, 29).

**Image segmentation, feature extraction, and data cleaning**

Batch image segmentation and feature extraction were performed using CellProfiler (version 2.1.1 rev6c2d896) (67), as previously described (17). Briefly, for ACFM images, EEF objects were segmented using a global Otsu thresholding strategy of the PbHSP70 image. Within each image set, the EEF object was either shrunk or expanded by two pixels for subsequent image masking to ensure the exclusion of HepG2-associated signal in masked OPP-A555 images used for intensity feature extraction. DNA objects within EEF objects were also identified by global thresholding and unified based on known parasite sizes. All HepG2 nuclei identified in an image were unified into a single object, and the OPP-A555 fluorescence intensity features of this unified object were used to quantify specific signals associated with HepG2 translation. Size, shape, and fluorescence intensity features of both EEFs and in-image HepG2 were exported from CellProfiler for downstream analysis using KNIME (68). To ensure the analysis included data from ACFM image sets containing one (and only one) true EEF in a HepG2 monolayer, as opposed to a piece of fluorescent debris or a dying cell not integrated in the monolayer, we computationally removed all data associated with image sets in which more than one EEF object was identified, the EEF object identified did not contain a DNA signal, and no HepG2 nuclei were identified. In addition, out-of-focus EEFs were identified and removed using object location filters in X and Y planes. Finally, metrics for EEF object area, EEF object circularity (compactness), and the identification of unified EEF DNA objects were used to identify cases of segmentation failures in which two parasites were segmented as a single EEF object. Our workflow for quality control of ACFM images can be accessed at https://hub.knime.com/-/spaces/-/~TZCrKvv3sbJwM_xP/current-state/.

**Merosome/detached cell collection and counting**

For merosome/detached cell collections in Figs 4 and 6, 1 ml iDMEM from each infected well was collected and transferred to a 24-well plate; then, 500 $\mu$l of iDMEM was gently added to each infected well, then collected and pooled with the initial 1 ml; and the merosomes/detached cells were allowed to settle for 15 min

before being counted using a light microscope equipped with an X-Y adjustable stage at 40x magnification. For collections in Figs 4B and 6D–J, 1 ml of iDMEM was added back to each infected monolayer and incubated for an additional 24 h, with merosome/detached cell recovery and counting was repeated at the 96- and 120-h timepoints.

### Data analysis, concentration–response curve fitting, and statistics

For concentration–response analysis, four-parameter non-linear regression curve fitting was performed in GraphPad Prism (version 7.0d) with the minimal response (top of the curve) set at 100, a constrained hill slope (–10< hill slope< 0). For analyses in which maximal effect was reached with ≥2 consecutive concentrations, the maximal effect was fit open; if no such plateau was achieved, the curve was fit with maximal effect constrained to 0. $EC_{50}$ and 95% CI were determined for each compound from ≥3 independent experiments. One-way ANOVA and ordinary two-way ANOVA with Tukey's multiple comparisons post hoc testing were performed in GraphPad Prism (version 9.5.1) using the mean of three independent experiments. For the ordinary two-way ANOVA and Tukey's post hoc tests, variation was analyzed by timepoint, compound, and the interaction of the two variables with independent analyses performed on 5x and 10x $EC_{50}$ concentrations. Test results for each ANOVA and associated post hoc tests are detailed in Table S2. All other data and statistical analyses were performed in KNIME (version 4.5.1). In Fig 3C, we modeled % growth per hour, with the assumption that growth would be constant throughout the time period analyzed, by determining the percent growth (experiment mean EEF area) from one timepoint to another, divided by the number of hours separating the two measurements; for example, (((28 hpi area – 24 hpi area)/24 hpi area)*100)/4 would yield % growth per hour between 24 and 28 hpi. Plots were constructed using KNIME, Plotly Chart Studio, and the SuperPlotsOfData online tool (https://huygens.science.uva.nl/SuperPlotsOfData/).

## Data Availability

In addition to the supplementary tables, the complete dataset used to generate Fig 1A and B can be downloaded and explored via interactive dashboards via our KNIME hub workflow at https://hub.knime.com/-/spaces/-/~TZCrKvv3sbJwM_xP/current-state/. The complete dataset used to generate Figs 2 and 3 and associated supplements can be downloaded and explored via interactive dashboards via our KNIME hub workflow at https://hub.knime.com/-/spaces/-/~EcnvMwYtqylu2reV/current-state/. ACFM raw image datasets are available on BioImage Archive (Accession ID: S-BIAD1040).

## Supplementary Information

## Acknowledgements

This work was supported by the National Institutes of Health grant R21AI149275 to KK Hanson. We thank the University of Georgia SporoCore and New York University Insectary for providing *P. berghei*-infected mosquitos. We thank William Sausman, Francisco Medrano, Beatriz Morales-Hernandez, and Daniel Ferguson for assistance with live luciferase assays.

### Author Contributions

JL McLellan: conceptualization, data curation, formal analysis, investigation, visualization, and writing—original draft, review, and editing.
KK Hanson: conceptualization, formal analysis, supervision, funding acquisition, investigation, visualization, methodology, project administration, and writing—original draft, review, and editing.

### Conflict of Interest Statement

The authors declare that they have no conflict of interest.

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
