## [Reviewer comments · Life Science Alliance]

Life Science Alliance

Differential effects of translation inhibitors on *P. berghei* liver stage parasites

James McLellan and Kirsten Hanson

DOI: <https://doi.org/10.26508/lsa.202302540>

Corresponding author(s): Kirsten Hanson, The University of Texas at San Antonio

Review Timeline:

Submission Date:	2023-12-18
Editorial Decision:	2024-02-05
Revision Received:	2024-03-07
Editorial Decision:	2024-03-11
Revision Received:	2024-03-19
Accepted:	2024-03-19

Transaction Report:

February 5, 2024

Re: Life Science Alliance manuscript #LSA-2023-02540

Kirsten K. Hanson
University of Texas at San Antonio

Dear Dr. Hanson,

Thank you for submitting your manuscript entitled "Translation inhibition efficacy does not determine the Plasmodium berghei liver stage antiplasmodial efficacy of protein synthesis inhibitors" to Life Science Alliance. The manuscript was assessed by expert reviewers, whose comments are appended to this letter. We invite you to submit a revised manuscript addressing the Reviewer comments.

Thank you for this interesting contribution to Life Science Alliance. We are looking forward to receiving your revised manuscript.

Sincerely,

B. MANUSCRIPT ORGANIZATION AND FORMATTING:

Reviewer #1 (Comments to the Authors (Required)):

The authors measured o-propargyl puromycin (OPP) mean intensity fluorescent (OPP_MFI) to quantify translation activity as well as the sizes of exoerythrocytic forms (EEFs) and the ability of EEFs to release hepatic merozoites (growth) after drug washout to evaluate antiplasmodial effects of five compounds. The major conclusion was that translation inhibition efficacy cannot function as a proxy for antiplasmodial effectiveness. The observation is interesting but is not unexpected. These compounds can have some unknown off-target effects. Indeed, their data clearly showed that DDD107498 had a non-translational inhibition effect in killing the parasites. Correlations between translation inhibition efficacy and antiplasmodial effects of the compounds were also performed, and correlations were found to be compound-specific. These data were from in vitro experiments, and the in vivo effects could be different.

Although the authors did many experiments measuring and collecting the data, the work is descriptive. It would be more interesting if they could focus on one or two compounds and dissect the mechanisms of observations or the off-target effects. As an antimalarial drug, why do we need to use the translation inhibition index to evaluate a drug? Directly testing drug inhibition or killing capability is also quite simple and is more practical.

The paper can be shortened; there are repeats of the same messages in the introduction, results, and discussion. Similar/same data were represented in multiple ways (Figures and supplemental figures). The discussion is long; some are not relevant.

Some minor points:

Figure S1A, left panel: spell out SPX, OPP, and ACFM. The legend for B and C, Single parasite translational intensity (OPP-MFI): should be 'Single parasite OPP-MFI'?

Line 164-193, mostly these are methods.

Figure 3, some statistical tests may help.

Figure S5, GFP should be MSP1? Please explain in the legend or change the label.

In Figure 4B, there are three colors in the bars. I assume they represent counts from three different time points.

Line 719-723, the secondary antibodies should be: "Alexa labeled anti-rabbit/rat/donkey IgG"?

Ref 21, 31, 36, 41, and 52 missing page number. Ref 66, no title.

Figure 1C, the triangle at 24h has a different color.

Reviewer #2 (Comments to the Authors (Required)):

McLellan and Hanson employ a *Plasmodium berghei*-HepG2 infection model to assess whether, and to what extent, inhibition of protein synthesis by five inhibitors of protein translation correlates with their antiplasmodial activity during the hepatic stage of infection by malaria parasites. Through a combination of proteome labelling and confocal microscopy, they assessed the translation inhibition capacity of the compounds in question and compared it with their liver stage antiplasmodial activity. The authors conclude that translation inhibition efficacy is not an accurate proxy for a translation inhibitor's antiplasmodial effectiveness, suggesting that the understanding of their mechanistic action on liver stage parasites requires further investigation.

The topic is relevant, the manuscript is well written, and the conclusions are generally supported by the data. I have only a few issues that the authors should address before the manuscript is published.

Abstract, lines 9-13 constitute a single, long sentence. Consider splitting in two.

Abstract, lines 29-31, refer compound DDD107498's "antimalarial effects". This is inaccurate and misleading, as the authors are measuring the compound's in vitro activity against *Plasmodium* hepatic infection, not malaria. Even if the compound was being tested for inhibition of *P. falciparum* growth in vitro, the term "antimalarial" would still be incorrect, as there is no malaria in an in vitro system. Thus, the term "antiplasmodial" should be employed instead.

Abstract, line 33, should be "compounds" instead of "compound".

Introduction, line 62. There is no such thing as "multistage antimalarial activity". A compound can have multistage antiplasmodial activity, but malaria only occurs during the blood stage of infection of the vertebrate host.

Introduction, lines 49-... The authors should refer to recent publications in ACS Infect. Dis. (PMIDs 31479238 and 35312290), where DDD107498's activity against *P. berghei* hepatic infection in vitro and in vivo are described.

Introduction, lines 93-98. Consider splitting the sentence in two and end with a brief summary of observations, as an Introduction should.

Results, lines 111-113. The sentence "Anisomycin (...) has similar activity against *P. berghei* LS translation and that of HepG2 cells" should be rephrased for clarity.

Results, line 166. Unclear what "4 treatment" refers to, please clarify.

Figure 1A. YY-axes labels must specify what is being measured. "% of control" does not provide enough information. Orange lines are missing in plots 3 and 5 of Figure 1A (at least in the version I had access to).

Figure 1C is not called in the text and only referred to in the legend of Figure 3. It must be called in the text. Also, it is unclear to me why the schematics of the experimental procedure behind the data in Figs. 1A and 1B should appear as the third, rather than the first panel of Figure 1.

Results, lines 182-189. "At 72 hpi, LS parasite maturation and hepatic merozoite formation were assessed by quantifying the percentage of monolayer EEFs expressing merozoite surface protein 1 (MSP1) and apical membrane antigen 1 (AMA1) in immunolabeled monolayers; we collected the culture medium from the same wells to quantify merozoite-filled, detached HepG2 cells, the presentation of merozoite formation in the *P. berghei*-HepG2 infection model (30). In parallel, independent samples were set up for repeat merozoite collection every 24 hours at 72-, 96- and 120 hpi, with complete medium replacement after each collection". It is unclear to me how "quantifying the PERCENTAGE of monolayer EEFs expressing MSP1 and AMA1" can inform on parasite growth/size. Representative images/plots of these analyses/measurements should be provided as supplementary material. Merely providing the data as "% of growth inhibition" on the YY-axes of plots does not give the reader an accurate notion of how these measurements were carried out.

Results, line 236. "is" should be "are".

Results, line 241. "4" should be "Four".

Results, lines 243-245. "respectively" missing after "tested".

Figure 3A. Can the authors explain why anisomycin-, LysRS-IN-2- and bruceantin-treated EEFs exceeded the DMSO control's translational intensity at 44 hpi. Could this be a recovery mechanism by the parasite to "compensate for lost time"?

Results, line 297. "release in from the monolayer" needs to be corrected.

Results, line 302 "begin" should be "begins".

Results, lines 309-310. "monolayers treated with anisomycin contained fewer hepatic merozoites (60-63% of control) compared to LysRS-IN-2 treatments (61-69% of control)". Given the overlapping intervals and standard deviations of these measurements, these differences are not statistically significant and, therefore, the statement that anisomycin-treated monolayers contained FEWER hepatic merozoites than their LysRS-IN-2-treated counterparts is incorrect and should not be made.

Results, line 353. "compound specific" should be "compound-specific".

Discussion absolutely must be significantly shortened.

References, line 834. Ref. 15 incomplete.

References, line 876. Ref. 34 needs re-formatting.

Reviewer #3 (Comments to the Authors (Required)):

McLellan and Hanson explore the relationship between translation inhibition efficacy and antiplasmodial effectiveness of protein synthesis inhibitors against *Plasmodium berghei* liver stage parasites. The authors investigate five mechanistically distinct compounds, including the leading antimalarial candidate DDD107498, using the *in vitro* *P. berghei*-HepG2 liver stage infection model. The paper addresses the crucial need for new antimalarial drugs that target various life cycle stages of *Plasmodium* parasites.

The manuscript is well-written, and the data presented in a clear and concise manner. The authors effectively highlight the importance of understanding the relationship between translation inhibition and antiplasmodial efficacy, especially in the context of liver stage parasites, which play a crucial role in the amplification of the parasite population. Importantly, the inclusion of

single parasite image sets and the analysis of translation inhibition EC50s for each compound contribute to the robustness of the study.

The data conclusively show that translation inhibition efficacy alone does not determine the antiplasmodial efficacy of the compounds. The authors effectively use DDD107498 as an example, demonstrating its strong antiplasmodial effects despite being the least effective translation inhibitor. The identification of compound-specific heterogeneity in parasite responses adds depth to the study and emphasizes the need for a nuanced understanding of the compounds' modes of action.

The authors appropriately interpret the results in the context of existing knowledge, addressing the demand for protein synthesis during liver stage development and the potential sensitivity of replicating liver stage parasites to translation inhibition. The authors acknowledge unexpected growth dynamics during the liver stage, highlighting the importance of exploring both proximate and ultimate mechanisms of action.

Thus, I have no doubts that this manuscript makes a valuable contribution to the field of antimalarial drug development by challenging the assumption that translation inhibition efficacy is a reliable proxy for antiplasmodial effectiveness. The study is well-designed, the results are clearly presented, and the implications of the findings are thoroughly discussed. Understanding the nuanced relationship between mode of action and antiplasmodial efficacy is crucial for identifying compounds with desirable activity profiles. Therefore, this paper deserves to be published, as it enhances our understanding of the mode of action of anti-malarial drugs and contributes to the ongoing efforts to combat malaria.

Reviewer #1 (Comments to the Authors (Required)):

We thank the Reviewer for their insight and constructive comments.

The authors measured o-propargyl puromycin (OPP) mean intensity fluorescent (OPP_MFI) to quantify translation activity as well as the sizes of exoerythrocytic forms (EEFs) and the ability of EEFs to release hepatic merozoites (growth) after drug washout to evaluate antiplasmodial effects of five compounds.

To clarify, we do not quantify hepatic merozoite formation or merosomes/detached cells to assess growth. These metrics assess late liver stage differentiation and maturation; a parasite which forms a merozoite-filled merosome/detached cell can be said to have successfully completed the liver stage *in vitro*. We measure parasite areas and extrapolate to growth based only on timepoints preceding late liver stage differentiation. We also want to stress that we did not do these experiments to evaluate the antiplasmodial effects of the compounds *per se*, but rather to uncover the relationship between translation inhibition efficacy and antiplasmodial efficacy specifically for compounds thought to act by blocking protein synthesis.

The major conclusion was that translation inhibition efficacy cannot function as a proxy for antiplasmodial effectiveness. The observation is interesting but is not unexpected. These compounds can have some unknown off-target effects. Indeed, their data clearly showed that DDD107498 had a non-translational inhibition effect in killing the parasites. Correlations between translation inhibition efficacy and antiplasmodial effects of the compounds were also performed, and correlations were found to be compound-specific. These data were from *in vitro* experiments, and the *in vivo* effects could be different. Although the authors did many experiments measuring and collecting the data, the work is descriptive. It would be more interesting if they could focus on one or two compounds and dissect the mechanisms of observations or the off-target effects. As an antimalarial drug, why do we need to use the translation inhibition index to evaluate a drug? Directly testing drug inhibition or killing capability is also quite simple and is more practical.

We designed these experiments to test a particular hypothesis: whether or not translation inhibition efficacy drives liver stage antiplasmodial efficacy *in vitro*, specifically for compounds that are known to act, mechanistically, by inhibiting protein synthesis regardless of molecular mechanism or which step in the process is inhibited. There are only two outcomes, and it could be rationally argued that either of the two would be not unexpected. However, before this work, the relationship was not known. We agree, of course, that *in vivo* pharmacology adds additional complexity, but the compounds progressed into *in vivo* studies, and even more labor intensive *in vitro* assays such as the PRR are usually selected on the basis of rapid and relatively easy *in vitro* assays. But the reason the PRR assay is quite valuable is that it can separate compounds which look identical in a canonical 48 or 72 hour *P. falciparum* ABS assay based on clinically desirable properties. Should a (reasonably) simple liver stage assay be able to do the same, specifically for compounds which target parasite translation, we believe it would indeed be useful. Unfortunately, quantification of translation inhibition efficacy clearly does not suffice. We absolutely agree that it will be interesting to attempt to tease out more mechanistic detail about ultimate killing mechanisms for compounds like DDD107498, now that our work has made clear

that the relationship between translation inhibition efficacy and liver stage antiplasmodial efficacy is not at all straightforward.

The paper can be shortened; there are repeats of the same messages in the introduction, results, and discussion.

We have gone through the paper and have shortened it. We have tried to remove repetitive prose where it was not important to the arguments being made, particularly in the discussion.

Similar/same data were represented in multiple ways (Figures and supplemental figures).

This is by design. We strongly favor presenting cumulative normalized data encompassing all individual parasites from all experiments when presenting translation intensity and size, as in Figs. 2, and 3A-B. Allowing the reader to visually assess reproducibility of these metrics between individual experiments is also important though, thus Fig. S3. We find the summation of all experimental data at the timepoints assessing effects of translation inhibitor treatment and subsequent recovery after washout on translation and growth (24-, 28-, 32-, and 44 hpi) in Fig. S5 extremely helpful in comparing compounds, but completely inadequate as a primary figure as no variability is captured.

The discussion is long; some are not relevant.

As mentioned above, we have gone through the whole paper and have shortened it. We have tried to remove repetitive prose where it was not important to the arguments being made, particularly in the discussion. We do not find that any of the topics we address in the discussion are irrelevant.

Some minor points:

Figure S1A, left panel: spell out SPX, OPP, and ACFM. The legend for B and C, Single parasite translational intensity (OPP-MFI): should be 'Single parasite OPP-MFI'?

We have added definitions of all abbreviations in Fig. S1A to the legend, and have used OPP-MFI for the Y axis labels in Fig. S1B-C. The in-figure legends of Fig. S1B-C show that different colors are used to represent which experiment a given parasite measurement is from.

Line 164-193, mostly these are methods.

We find it crucial that a reader understands the full experimental setup to assess the results, which is why we describe it and schematize it in Figure 1C.

Figure 3, some statistical tests may help.

We have performed ordinary two-way ANOVA with Tukey's multiple comparisons post hoc testing to assess the variance between compounds, timepoints, and the interaction between these two factors in the 5x- and 10x EC50 concentrations separately. We've added the results of these analyses to table S2 and the Results.

Figure S5, GFP should be MSP1? Please explain in the legend or change the label.

Figure S5 is labeled correctly; MSP1 is not shown in Fig. S5. MSP1 is a common marker for late liver stage maturation, and we cite papers show that the MSP1 is detected substantially before merozoites are formed in the text. While used extensively in *P. falciparum* asexual blood stage work, AMA1 is less common as a liver stage marker, so we provided this figure to demonstrate that AMA1 is not yet detectable in cytomere-stage parasites, but is present once hepatic merozoites are forming.

In Figure 4B, there are three colors in the bars. I assume they represent counts from three different time points.

We apologize for omitting the timepoint legend from this figure. This was corrected.

Line 719-723, the secondary antibodies should be: "Alexa labeled anti-rabbit/rat/donkey IgG"?

We have added in the "IgG" for all secondary antibodies in the Methods.

Ref 21, 31, 36, 41, and 52 missing page number. Ref 66, no title.

These references were updated.

Figure 1C, the triangle at 24h has a different color.

Both the triangle and the circle in the schematic at 24 hpi had a greater transparency than for subsequent timepoints as 24 hpi is only a single pre-treatment control sample. We have changed the colors to match those at 28, 32, and 44 hpi for simplicity.

Reviewer #2 (Comments to the Authors (Required)):

McLellan and Hanson employ a *Plasmodium berghei*-HepG2 infection model to assess whether, and to what extent, inhibition of protein synthesis by five inhibitors of protein translation correlates with their antiplasmodial activity during the hepatic stage of infection by malaria parasites. Through a combination of proteome labelling and confocal microscopy, they assessed the translation inhibition capacity of the compounds in question and compared it with their liver stage antiplasmodial activity. The authors conclude that translation inhibition efficacy is not an accurate proxy for a translation inhibitor's antiplasmodial effectiveness, suggesting that the understanding of their mechanistic action on liver stage parasites requires further investigation.

The topic is relevant, the manuscript is well written, and the conclusions are generally supported by the data. I have only a few issues that the authors should address before the manuscript is published.

We thank the Reviewer for their detailed assessment and suggestions to improve the clarity and accuracy of our work.

Abstract, lines 9-13 constitute a single, long sentence. Consider splitting in two.

The abstract was altered to meet the journal-specific constraints, so this has been changed.

Abstract, lines 29-31, refer compound DDD107498's "antimalarial effects". This is inaccurate and misleading, as the authors are measuring the compound's *in vitro* activity against Plasmodium hepatic infection, not malaria. Even if the compound was being tested for inhibition of *P. falciparum* growth *in vitro*, the term "antimalarial" would still be incorrect, as there is no malaria in an *in vitro* system. Thus, the term "antiplasmodial" should be employed instead.

We agree, and have tried to be precise in our use of "antiplasmodial" when describing *in vitro* experiments throughout the paper. The abstract has been substantially altered to comply with the journal-specific requirements, so this has been removed.

Abstract, line 33, should be "compounds" instead of "compound".

Corrected.

Introduction, line 62. There is no such thing as "multistage antimalarial activity". A compound can have multistage antiplasmodial activity, but malaria only occurs during the blood stage of infection of the vertebrate host.

We strongly disagree. There is such a thing as "multistage antimalarial activity"; multistage antimalarial compounds are foundational to current antimalarial drug discovery and development. Liver stage-active antimalarials are compounds which can prevent malaria, by killing parasites before they reach the blood stages of the life cycle. Obviously, an antimalarial compound that was exclusively active against liver stage parasites could only be used to prevent, but not treat malaria. The concept of multistage antimalarials is widespread in the literature, e.g. "Cytoplasmic isoleucyl tRNA synthetase as an attractive multistage antimalarial drug target" (DOI: [10.1126/scitranslmed.adc9249](https://doi.org/10.1126/scitranslmed.adc9249)), "Discovery of a Quinoline-4-carboxamide Derivative with a Novel Mechanism of Action, Multistage Antimalarial Activity, and Potent *In Vivo* Efficacy" (DOI: [10.1021/acs.jmedchem.6b00723](https://doi.org/10.1021/acs.jmedchem.6b00723)), and "Drug Screen Targeted at Plasmodium Liver Stages Identifies a Potent Multistage Antimalarial Drug" (DOI: [10.1093/infdis/jis184](https://doi.org/10.1093/infdis/jis184)).

Introduction, lines 49-... The authors should refer to recent publications in ACS Infect. Dis. (PMIDs 31479238 and 35312290), where DDD107498's activity against *P. berghei* hepatic infection *in vitro* and *in vivo* are described.

The sentence beginning in line 49 is specifically referring to DDD107498 (or cabamaquine) efficacy against *P. falciparum* asexual blood and liver stages in human volunteers, while the previous sentence is referring to the evidence that elongation factor 2 is the target of DDD107498 so we don't feel these references are relevant here, and do not want to expand our introduction, further lengthening the paper. We have incorporated the results from 25 nM DDD107498 treatment from 24-48 hpi in the HepG2 spheroid model into our discussion, though, and now cite that paper there.

Introduction, lines 93-98. Consider splitting the sentence in two and end with a brief summary of observations, as an Introduction should.

While we appreciate and understand the reviewer's point that many introductions include a brief summary of the data, others take the same approach as we do: setting out the key question to be addressed (e.g. DOI: 10.26508/lsa.202101237) .

Results, lines 111-113. The sentence "Anisomycin (...) has similar activity against *P. berghei* LS translation and that of HepG2 cells" should be rephrased for clarity.

This has been rephrased.

Results, line 166. Unclear what "4 treatment" refers to, please clarify.

This was an overlooked typo and was intended to say, "at the end of a 4-h treatment". We apologize for the confusion and have corrected the mistake in the manuscript.

Figure 1A. YY-axes labels must specify what is being measured. "% of control" does not provide enough information. Orange lines are missing in plots 3 and 5 of Figure 1A (at least in the version I had access to).

We have re-labeled the Y-axis to read "OPP MFI, % of control". Regarding the missing orange lines, no curve was fit for DDD107498 and MMV019266 activity against HepG2 translation as neither was active at even the highest concentration tested, so no curve is shown, only the data points. We have added a clarification about this to the Figure 1 legend to avoid any confusion.

Figure 1C is not called in the text and only referred to in the legend of Figure 3. It must be called in the text. Also, it is unclear to me why the schematics of the experimental procedure behind the data in Figs. 1A and 1B should appear as the third, rather than the first panel of Figure 1.

We mis-labeled the in-text reference to Figure 1C in the original manuscript and have corrected this. We apologize for this mistake and any resulting misunderstanding. Regarding the order of the subpanels, the schematic is located after Fig. 1A-B because it does not pertain to these concentration-response experiments. The schematic in Fig. 1C pertains to data presented in the subsequent figures (Figs 2-4 and related supplements) where we assess the effects of these compounds tested in equivalent effective concentrations (determined based on Fig. 1A-B). Further, each type of experimental endpoint shown in the schematic is color coded to match the plots in later figures, e.g. the experimental endpoints where parasite translation is assessed are marked by a red triangle in Fig. 1C and correspondingly, the figures pertaining to parasite translation are plotted in shades of red (Fig. 2A and Fig. 3A).

Results, lines 182-189. "At 72 hpi, LS parasite maturation and hepatic merozoite formation were assessed by quantifying the percentage of monolayer EEFs expressing merozoite surface protein 1 (MSP1) and apical membrane antigen 1 (AMA1) in immunolabeled monolayers; we collected the culture medium from the same wells to quantify merozoite-filled, detached HepG2 cells, the presentation of merozoite formation in the *P. berghei*-HepG2 infection model (30). In parallel, independent samples were set up for repeat merozoite collection every 24 hours at 72-, 96- and

120 hpi, with complete medium replacement after each collection". It is unclear to me how "quantifying the PERCENTAGE of monolayer EEFs expressing MSP1 and AMA1" can inform on parasite growth/size. Representative images/plots of these analyses/measurements should be provided as supplementary material. Merely providing the data as "% of growth inhibition" on the YY-axes of plots does not give the reader an accurate notion of how these measurements were carried out.

We apologize for this misunderstanding, which we suspect may be due to our in-text error in reference to Figure 1C. For clarification, the percentage of monolayer EEFs expressing MSP1 and AMA1 was used as a metric to assess parasite maturation and hepatic merozoite formation (reported in Figure 4A), not parasite size or growth. Parasite area is measured at 24-, 28-, 32-, and 44 hpi (as indicated in Fig. 1C), and we model growth per hour from 24-28, 28-32, and 32-44 according to the formula provided in the Methods. With respect to "% of growth inhibition", we suspect you are referring to Fig. 2 and Fig. S2. For these, both translation (OPP MFI) and parasite area were measured and normalized on a per-experiment basis to the DMSO control mean, which was set to 100. We apologize for omitting this important information. It is now present in the legends for Fig. 2 and Fig. S2. We note that the full non-normalized dataset of size and translation measurements (at 24-, 28-, 32, and 44- hpi) is available via our interactive workflow on the KNIME hub (cited in the paper) and is further provided with our resubmission as "source data".

Results, line 236. "is" should be "are".

Results, line 241. "4" should be "Four".

Results, lines 243-245. "respectively" missing after "tested".

These have been corrected.

Figure 3A. Can the authors explain why anisomycin-, LysRS-IN-2- and bruceantin-treated EEFs exceeded the DMSO control's translational intensity at 44 hpi. Could this be a recovery mechanism by the parasite to "compensate for lost time"?

We can only speculate. Average parasite translation intensity drops profoundly between 24 and 48 hpi (Fig. S1A), and this decrease in OPP-MFI is also clear at 44 hpi (not shown). We interpret the greater mean average translational intensity of these parasite populations relative to the control as a result of them being developmentally delayed, again relative to the control. The reviewer's suggested interpretation is also an intriguing one.

Results, line 297. "release in from the monolayer" needs to be corrected.

Results, line 302 "begin" should be "begins".

These have been corrected.

Results, lines 309-310. "monolayers treated with anisomycin contained fewer hepatic merozoites (60-63% of control) compared to LysRS-IN-2 treatments (61-69% of control)". Given the overlapping intervals and standard deviations of these measurements, these differences are not

statistically significant and, therefore, the statement that anisomycin-treated monolayers contained FEWER hepatic merozoites than their LysRS-IN-2-treated counterparts is incorrect and should not be made.

We agree with the reviewer and have removed this statement.

Results, line 353. "compound specific" should be "compound-specific".

This has been corrected.

Discussion absolutely must be significantly shortened.

We have gone through the whole paper and have shortened it. We have tried to remove repetitive prose where it was not important to the arguments being made, particularly in the discussion.

References, line 834. Ref. 15 incomplete.

References, line 876. Ref. 34 needs re-formatting.

We thank the reviewer for pointing out these formatting errors. When importing citations from certain sources, the reference manager is incapable of populating certain fields. These references were manually corrected.

Reviewer #3 (Comments to the Authors (Required)):

We thank the Reviewer for their effort reviewing our work and appreciate the detailed assessment.

McLellan and Hanson explore the relationship between translation inhibition efficacy and antiplasmodial effectiveness of protein synthesis inhibitors against *Plasmodium berghei* liver stage parasites. The authors investigate five mechanistically distinct compounds, including the leading antimalarial candidate DDD107498, using the in vitro *P. berghei*-HepG2 liver stage infection model. The paper addresses the crucial need for new antimalarial drugs that target various life cycle stages of *Plasmodium* parasites.

The manuscript is well-written, and the data presented in a clear and concise manner. The authors effectively highlight the importance of understanding the relationship between translation inhibition and antiplasmodial efficacy, especially in the context of liver stage parasites, which play a crucial role in the amplification of the parasite population. Importantly, the inclusion of single parasite image sets and the analysis of translation inhibition EC50s for each compound contribute to the robustness of the study.

The data conclusively show that translation inhibition efficacy alone does not determine the antiplasmodial efficacy of the compounds. The authors effectively use DDD107498 as an example, demonstrating its strong antiplasmodial effects despite being the least effective translation inhibitor. The identification of compound-specific heterogeneity in parasite responses

adds depth to the study and emphasizes the need for a nuanced understanding of the compounds' modes of action.

The authors appropriately interpret the results in the context of existing knowledge, addressing the demand for protein synthesis during liver stage development and the potential sensitivity of replicating liver stage parasites to translation inhibition. The authors acknowledge unexpected growth dynamics during the liver stage, highlighting the importance of exploring both proximate and ultimate mechanisms of action.

Thus, I have no doubts that this manuscript makes a valuable contribution to the field of antimalarial drug development by challenging the assumption that translation inhibition efficacy is a reliable proxy for antiplasmodial effectiveness. The study is well-designed, the results are clearly presented, and the implications of the findings are thoroughly discussed. Understanding the nuanced relationship between mode of action and antiplasmodial efficacy is crucial for identifying compounds with desirable activity profiles. Therefore, this paper deserves to be published, as it enhances our understanding of the mode of action of anti-malarial drugs and contributes to the ongoing efforts to combat malaria.

March 11, 2024

RE: Life Science Alliance Manuscript #LSA-2023-02540R

Dr. Kirsten K. Hanson
The University of Texas at San Antonio
1 UTSA Circle
San Antonio, Texas 78249

Dear Dr. Hanson,

Thank you for submitting your revised manuscript entitled "Differential effects of translation inhibitors on *P. berghei* liver stage parasites". We would be happy to publish your paper in Life Science Alliance pending final revisions necessary to meet our formatting guidelines.

- please be sure that the authorship listing and order is correct
- please add ORCID ID for the corresponding author -- you should have received instructions on how to do so
- please add the Twitter handle of your host institute/organization as well as your own or/and one of the authors in our system
- please use the [10 author names et al.] format in your references (i.e., limit the author names to the first 10) and label the section as References
- please add callouts for Figures S3A-B and S7A-B to your main manuscript text
- please add a conflict of interest statement to your main manuscript text

A. FINAL FILES:

B. MANUSCRIPT ORGANIZATION AND FORMATTING:

**Submission of a paper that does not conform to Life Science Alliance guidelines will delay the acceptance of your

manuscript.**

The license to publish form must be signed before your manuscript can be sent to production. A link to the electronic license to publish form will be available to the corresponding author only. Please take a moment to check your funder requirements.

Sincerely,

March 19, 2024

RE: Life Science Alliance Manuscript #LSA-2023-02540RR

Dr. Kirsten K. Hanson
The University of Texas at San Antonio
1 UTSA Circle
San Antonio, Texas 78249

Dear Dr. Hanson,

Thank you for submitting your Research Article entitled "Differential effects of translation inhibitors on *P. berghei* liver stage parasites". It is a pleasure to let you know that your manuscript is now accepted for publication in Life Science Alliance. Congratulations on this interesting work.

DISTRIBUTION OF MATERIALS:

Again, congratulations on a very nice paper. I hope you found the review process to be constructive and are pleased with how the manuscript was handled editorially. We look forward to future exciting submissions from your lab.

Sincerely,
